# Image Stitching in Adverse Condition: A Bidirectional-Consistency Learning Framework and Benchmark

**Zengxi Zhang**
The University of Tokyo
cyouzoukyuu@gmail.com

**Junchen Ge**
Tsinghua University
junchen54ge@gmail.com

**Zhiying Jiang**
Dalian Martime University
zyjiang0630@gmail.com

**Miao Zhang**
Dalian University of Technology
miaozhang@dlut.edu.cn

**Jinyuan Liu**\*
Dalian University of Technology
atlantis918@hotmail.com

## Abstract

Deep learning-based image stitching methods have achieved promising performance on conventional stitching datasets. However, real-world scenarios may introduce challenges such as complex weather conditions, illumination variations, and dynamic scene motion, which severely degrade image quality and lead to significant misalignment in stitching results. To solve this problem, we propose an adverse condition-tolerant image stitching network, dubbed ACDIS. We first introduce a bidirectional consistency learning framework, which ensures reliable alignment through an iterative optimization paradigm that integrates differentiable image restoration and Gaussian-distribute encoded homography estimation. Subsequently, we incorporate motion constraints into the seamless composition network to produce robust stitching results without interference from moving scenes. We further propose the first adverse scene image stitching dataset, which covers diverse parallax and scenes under low-light, haze, and underwater environments. Extensive experiments show that the proposed method can generate visually pleasing stitched images under adverse conditions, outperforming state-of-the-art methods. Code and benchmark are available at https://github.com/ZengxiZhang/ACDIS.

## 1 Introduction

Image stitching aims to construct a wide field-of-view (FoV) scene from multiple images captured in different viewpoints. It is widely used in various applications such as autonomous driving [1], map construction [2] and virtual reality [3]. Although it has achieved rapid development in recent years, aligning images over large disparity views, visually degraded environments and dynamic scenes still remains challenging.

Early stitching methods [4, 5, 6] predominantly relied on SIFT-based feature detection [7] to estimate parametric warping models. However, they fall short in low-texture regions or scenes with repetitive geometric patterns. In recent years, learning-based methods [8, 9, 10] replace handcrafted feature

---

\*Corresponding Author.

39th Conference on Neural Information Processing Systems (NeurIPS 2025).

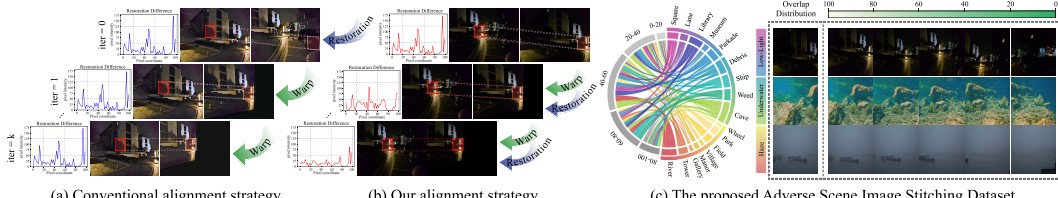

(a) Conventional alignment strategy     (b) Our alignment strategy     (c) The proposed Adverse Scene Image Stitching Dataset

Figure 1: (a) and (b) compare the alignment process of the traditional approach and our proposed strategy in challenging environments. The progression from top to bottom illustrates the refinement of alignment from coarse to fine, where the quadrilateral boxes denote regions corresponding to the same scene. The line charts on the left depict the current intensity difference of the restoration effect within the respective boxes. (c) presents the baseline and scene distribution of the proposed dataset.

extraction with deep semantic representations, enabling more robust alignment. Nevertheless, they still struggle to achieve satisfactory results in adverse environments (e.g., low-light, haze, underwater, and dynamic scenes). These conditions corrupt the image pairs through noise, color casts, or scene variation, leading to information distortion, which in turn affects feature matching and reconstruction.

To address this issue, some methods [11, 12, 13] employ image stitching by introducing environment-insensitive multi-modal data to alleviate the impact of degradation factors. However, these cannot be applied in general scenarios due to their strong data dependency. Furthermore, few methods mitigate image degradation in adverse environments without introducing additional information. On the other hand, existing stitching datasets [14, 9] are mainly collected under ideal natural light conditions and lack guiding reference, making it difficult to evaluate stitching tasks in degraded scenes.

To overcome the above limitations, in this paper, we propose a robust deep image stitching network for adverse conditions (ACDIS), which consists of two stages: In the first stage, we introduce a bidirectional-consistency learning framework to achieve robust image alignment under harsh environment. Specifically, considering that conventional homography estimation methods with deterministic displacement outputs often produce unreliable predictions in textureless or blurry regions, we begin by proposing a recursive parameterized homography estimation module that models displacements in a Gaussian-distributed transformation space. By jointly predicting the mean and variance, the model explicitly encodes uncertainty and mitigates overconfident regression. A Jensen–Shannon divergence–based optimization further refines the displacement distribution, yielding stable and reliable homography estimation.

Traditional perception methods in adverse conditions typically treat visual enhancement [15, 16, 17, 18] as a preprocessing step, as illustrated in Fig. 1 (a). However, as depicted in the line chart, this straightforward concatenation strategy may introduce irreducible restoration discrepancies, ultimately disrupting feature matching. To address this, as depicted in Fig. 1 (b), we embed a differentiable restoration module within the iterative homography estimation pipeline, enabling bidirectional optimization of both restoration and alignment. As shown in the corresponding line chart, this mechanism progressively harmonizes the scene structure, enhancing restoration consistency in iterative deformation processes, thereby achieving more reliable alignment.

In the second stage, we propose a motion-tolerant seamless composition network, which introduces motion loss to reduce the impact of moving object displacement caused by inconsistent image capturing times while generating clear stitched images. For evaluating the performance of stitching methods in harsh environments, we additionally propose the first adverse scene image stitching dataset (ASIS), which covers three harsh environments: low-light, haze, and underwater. ASIS includes 17 scenes with a total of 2,250 pairs of images, which are derived from manual capturing and network collection. All the image pairs are far from the plane structure and under a wide parallax range. In addition, we pre-align all image pairs to obtain the reference homography for a more comprehensive evaluation. In summary, our contributions are as follows:

- A recursive parameterized homography estimation module is proposed that iteratively encodes displacement within a Gaussian-distributed transformation space, progressively refining a reliable homography transformation.

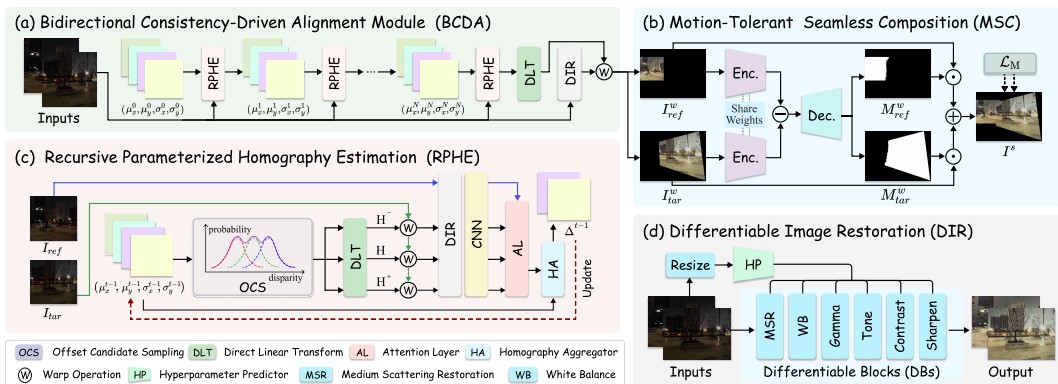

Figure 2: Workflow of the proposed Method.

- We propose a bidirectional-consistency learning framework, which achieves adverse condition-tolerant homography estimation by embedding lightweight differentiable restoration blocks into iterative alignment process.

- A motion-tolerant seamless composition network is introduced that generates visually pleasing stitching images while avoiding the interference of moving objects on temporally inconsistent image pairs.

- We release the first real-world adverse environment image stitching dataset, which contains 2,250 degraded image pairs with homography reference to evaluate the effectiveness of stitching methods in adverse conditions.

## 2 Related Work

### 2.1 Image Stitching Methods

Early traditional stitching works [4, 5, 6, 14, 19, 20, 21, 22] constructed global adaptive warping by using SIFT [7] to extract key points. Brown *et al.* [4] used multi-image matching technology in image stitching tasks and combined multi-band mixing to generate panoramas. Zaragoza *et al.* [14] fine-tuned the warping projection by moving direct linear transformations to reduce artifacts during warping, thereby reducing the reliance on ghost concealment algorithms. Peng *et al.* [23] introduced edge detection and sampled edges to construct triangles representing geometric structures. Then, a similarity transformation is performed by combining the geometric structure preserving energy term.

In recent years, some works have applied deep learning networks [8, 9, 11, 24, 12, 10] to the stitching task. Nie *et al.* [9] proposed an unsupervised image stitching method and introduced seam loss during the reconstruction process to eliminate the pixel-level misalignment. Then, they [10] employed a seam-driven mask as an alternative, effectively preserving structural coherence across the stitched regions. Although these methods have proven effective, in practice, immeasurable environmental factors often affect the captured images, thus greatly affecting the stitching effect.

### 2.2 Image Stitching Datasets

In recent years, many datasets [5, 21, 14, 19, 25] have been proposed to more comprehensively evaluate the effect of image stitching. Nie *et al.* [8] synthesized a stitching dataset with ground-truth by cropping and warping existing image datasets. Subsequently, considering the lack of disparity in synthetic datasets, they [9] additionally provided a real-world unsupervised dataset UDIS-D. However, as the source data are captured instead of synthesized, the ground truth of corresponding FoV scenes cannot be obtained, limiting quantitative evaluation. Kweon *et al.* [26] then proposed a stitching dataset with pixel-level warping and ground-truth stitching results, which simulated the real-world stitching by collected data from 3D virtual scenes. However, except for a small number of low-light images provided by UDIS-D, there is currently no stitching dataset specifically used to evaluate the robustness of the image stitching network in harsh environments.

# 3 The Proposed Method

Our method consists of two stages: Bidirectional Consistency-Driven Alignment Module (BCDA) and Motion-Tolerant Seamless Composition (MSC), as shown in Fig. 2. $\{I_{ref}, I_{tar}\}$ denotes images captured in adverse environments, where $ref$ and $tar$ represent the reference and target viewpoint.

BCDA comprises Recursive Parameterized Homography Estimation Module (RPHE) and Differentiable Image Restoration (DIR). RPHE alleviates the challenge of spatially limited dynamic cost volume with large baseline image pairs by sampling offset candidate values by mean and variance in 2D directions. DIR is designed to mitigate the interference of degradation factors on feature extraction. It applies CNN-encoded hyperparameters on differentiable restoration blocks to achieve lightweight and robust image restoration. Rather than applying the restoration module as a preprocessing step for alignment, we establish a Bidirectional-Consistency Learning Framework (BCLF), where DIR is embedded within the coarse-to-fine deformation estimation process, forming a progressive bidirectional optimization pipeline. Ultimately, BCDA achieves region-asymptotically consistent image pair restoration and robust image alignment. After obtaining $\{I_{ref}^w, I_{tar}^w\}$ by warping restored images, we input them into MSC to generate motion-tolerant wide field-of-view scene $I^s$. Next, we provide detailed information about each module.

## 3.1 Recursive Parameterized Homography Estimation

Some traditional stitching methods [9, 27, 28, 29] apply dynamic cost volume $\mathcal{C}(x) = c_{x,y}^r$ with search redius $r$ to match difference between $(x, y)$ of $I_{ref}$ and $(x \pm r, y \pm r)$ of $I_{tar}$ for reducing the memory consumption. However, spatial limited cost volume is not conducive to convergence in large baseline scenarios. Furthermore, these methods based on deterministic displacement outputs tend to produce unreliable predictions in textureless or blurry regions, and perform poorly in image stitching tasks under adverse conditions such as heavy fog or low light.

To address this, modeling displacement space $d$ as a Gaussian distribution provides a more robust alternative. The proposed model predicts not only the mean (displacement) but also the variance (as an uncertainty estimate [30]), allowing it to express high uncertainty in ambiguous or ill-posed regions and avoid overconfident regression to potentially incorrect values. Specifically, the cost volume will sample candidate offsets by mean $\mu$ and standard deviation $\sigma$ from each offset:

$$\mathcal{C}(i, \mu_i, \sigma_i) = c_i^{d(\mu_i, \sigma_i)}, \ d(\mu_i, \sigma_i) \sim \mathcal{N}\left(\mu_i, \sigma_i^2\right), \tag{1}$$

where $\sim$ represents the sampling from Gaussian distribution $\mathcal{N}$. $i \in \{x, y\}$, $x$ and $y$ denotes the horizontal and vertical displacements.

As shown in Fig. 2, we first sampling candidate offsets $\mu_i$, $\mu_i + \sigma_i$ and $\mu_i - \sigma_i$ from the initialized offsets in each iteration. Then, we transform them all into the homography $H$, $H^+$ and $H^-$ via Direct Linear Transformation (DLT) for warping the original images respectively. Subsequently, we input the warped images into the DIR (3.2) and pyramid CNN to extract features resilient to adverse environments. Through Attention Layer (AL) [31] and Homography Aggregation Module (HA) [31], we finally obtain the residual $\mu$ for updating. The detailed structure of CNN, AL and HA is illustrated in supplementary materials.

In order to effectively learn the parameters $\theta = \{\mu, \sigma\}$ of the Gaussian distribution, we use JS divergence [32] based optimization $\mathcal{J}$, which forces $\theta$ to gradually approach the Gaussian distribution during the optimization process:

$$\mathcal{J}_i = \frac{1}{2}\left(F\left(\mathcal{N}_{\mathrm{GT}} \| \mathcal{N}_i\right) + F\left(\mathcal{N}_i \| \mathcal{N}_{\mathrm{GT}}\right)\right), \tag{2}$$

where $\mathcal{N}_i$ is the short form of $\mathcal{N}\left(\mu_i, \sigma_i^2\right)$. $F\left(\cdot \| \cdot\right)$ represents KL divergence [33]. $\mathcal{N}_{\mathrm{GT}} = \mathcal{N}\left(\mu_{\mathrm{GT}}, \sigma_{\mathrm{GT}}^2\right)$ denotes predefined Gaussian distribution, where $\mu_{\mathrm{GT}}$ represents the ground truth offset. We can then update $\theta$ via feedforward gradient optimization [34, 35] in each iteration $t$:

$$
\begin{aligned}
\sigma_i^t &= \sigma_i^{t-1} - \frac{1}{2}\left(\frac{(\sigma_i^{t-1})^2 - \sigma_{\mathrm{GT}}^2 - (\mu_{\mathrm{GT}} - \mu_i^{t-1})^2}{(\sigma_i^{t-1})^3} - \frac{1}{\sigma_i^{t-1}} + \frac{\sigma_i^{t-1}}{\sigma_{\mathrm{GT}}^2}\right), \\
\mu_i^t &= \mu_i^{t-1} - \left(-\frac{\mu_{\mathrm{GT}} - \mu_i^{t-1}}{2}\left(\frac{1}{(\sigma_i^{t-1})^2} + \frac{1}{\sigma_{\mathrm{GT}}^2}\right)\right).
\end{aligned}
\tag{3}
$$

Considering that $\mu_{\mathrm{GT}}$ is not available during inference, we replaced the $\mu_{\mathrm{GT}}$ by approximate the optimizing step $\Delta^{t-1} = \mu_{\mathrm{GT}} - \mu_i^{t-1}$, which is estimated in each iteration. By updating $\theta$ in $N$ iterations, we can finally estimate a robust homography that is suitable for large baseline scenes in harsh conditions. L1 loss is used during the training stage with the ground truth of the offset in $N$ iterations, which can be described as:

$$L_1 = \sum_{t=1}^{N} \lambda_1^{(N-t)} \left| \mu^t - \mu_{\mathrm{GT}} \right|. \tag{4}$$

## 3.2 Bidirectional-Consistency Learning Framework

When conducting computer vision tasks in adverse conditions, previous works [36, 37, 38] usually directly employ restoration networks as the preprocessing step for the current task. However, when faced with stitching tasks, restoration networks encounter two main challenges: (1) Data-driven networks often overlook the physical imaging priors of images under adverse scenes, resulting in visually insensitive artifacts that implicitly disrupt the geometry structure of the images. (2) Restoration networks may exhibit varying degrees of effectiveness in restoring images from different perspectives. Both of these challenges can affect the feature matching between image pairs, thereby influencing warping estimation.

To address the first challenge, we propose a Differentiable Image Restoration Module (DIR), which includes a Hyperparameter Predictor (HP) and Differentiable Restoration Blocks (DBs). First, HP learns the global information of the resized input image to obtain hyperparameters. Then, we used the obtained hyperparameters as weights in DBs to achieve adaptive image restoration.

DBs consists of Medium Scattering Restoration (MSR), White Balance (WB), Gamma, Hue, Contrast and Sharpen. MSR estimates the atmospheric light $A$ and the transmission map $t(x)$ through the atmospheric light imaging model [39] to restore the absorption and scattering of light in the gas or liquid medium, which is expressed as $I(x) = J(x)t(x) + A(1 - t(x))$, where $I$ and $J$ denotes the degraded image and its clear counterpart. $A$ can be obtained by calculating the first 1000 brightest pixels in the dark channel and averaging the pixels at the corresponding positions of the original image. $t(x)$ can be obtained according to the Beer-Lambert law [40, 41], which is described as:

$$t(x, \omega) = 1 - \omega \min_{C} \left( \min_{y \in \Omega(x)} \frac{I^C(y)}{A^C} \right), \tag{5}$$

where $\omega$ is an hyperparameter that controls the degree of restoration, which is obtained by HP.

WB and Gamma are pixel-level restoration blocks. Among them, $\{\omega_r, \omega_g, \omega_b\}$ are used as hyperparameters of HP for mapping of three channel pixels. $G$ is the hyperparameter of Gamma for the weight of the power function. Tone restoration can be expressed as a channel-independent monotonic piecewise linear function [42]. The point $(k/M, T_k/T_M)$ on the tone curve is represented by M hyperparameters $\{t_0, ..., t_{M-1}\}$, where $T_k = \sum_{i=0}^{k-1} t_i$. The mapping process can be expressed as $I_o = \frac{1}{T_M} \sum_{j=0}^{M-1} \mathrm{clip} \left( L \cdot I_i - j, 0, 1 \right) t_k$, where $I_o$ and $I_i$ represent the output and the input image. Contrast restoration determines the linear difference between the $I_o$ and $I_i$ through the hyperparameter $\alpha$, which is expressed as $I_o = \alpha \cdot \mathrm{En} \left( I_i \right) + (1 - \alpha) \cdot I_i$, where En is the mapping process [42]. The sharpening [43] process is described as $F(x, \lambda) = I(x) + \lambda(I(x) - G(I(x)))$, where $G$ represents Gaussian filter. The hyperparameter $\lambda$ is a positive scaling factor to control the prominence of details.

In summary, different from traditional data-driven networks, DIR leverages physical priors to adaptively restore images in various adverse environments while minimizing interference, ensuring more accurate feature matching. We use L1 loss $\mathcal{L}_1$ and SSIM loss $\mathcal{L}_s$ with the ground truth image $I_{\mathrm{GT}}$ in the training process, which can be described as:

$$\mathcal{L}_{\mathrm{res}} = \lambda_2 \mathcal{L}_1(I_o, I_{\mathrm{GT}}) + \lambda_3 \mathcal{L}_s(I_o, I_{\mathrm{GT}}). \tag{6}$$

Although artifacts can be reduced with physical prior, consistent restoration of images from different perspective still remains challenging. Therefore, we further build a Bidirectional-Consistency Learning Framework. Specifically, we use DIR to restore the warped images at each iteration, rather than just once as a preprocessing means only before homography estimation process, achieving in equivariant restoration. This framework can be expressed as:

$$E_{tar}^n = \Phi \left( \mathcal{W} \left( I_{tar}; \hat{\mathbf{H}}^n \right) \right), \hat{\mathbf{H}}^{n+1} = \Psi \left( \mathbf{I}_{ref}; E_{tar}^n \right), \tag{7}$$

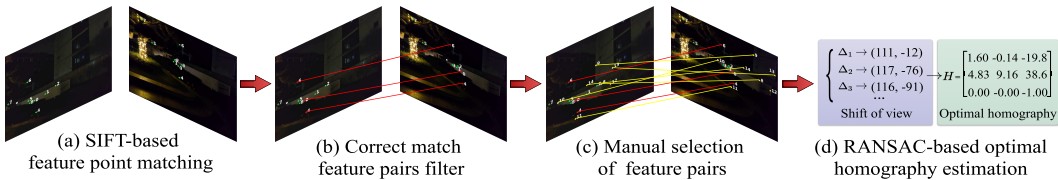

| (a) SIFT-based feature point matching | (b) Correct match feature pairs filter | (c) Manual selection of feature pairs | (d) RANSAC-based optimal homography estimation |

Figure 4: Reference generation process of the newly proposed ASIS dataset.

where $\Phi$ and $\Psi$ denote the restoration and the homography estimation and module. $E_{tar}^n$ and $\mathbf{H}^n$ represent the enhanced warping images and current homography in the $n$-th iteration. Since the computation cost of DIR is only 0.07 GFlops at each restoration, it will not significantly increase the time consumption in embedding it in the iteration. As the iteration proceeds, $I_{tar}^w$ becomes more aligned with $I_{ref}$, enabling DIR to increasingly achieve consistent restoration of overlapping regions, which further improves homography estimation.

### 3.3 Motion-Tolerant Seamless Composition

Inspired by UDIS2 [10], this stage generates a soft mask with floating point numbers and synthesizes a seamless stitched image $I^s$ by $I^s = M_{ref}^w \times I_{ref}^w + M_{tar}^w \times I_{tar}^w$. U-Net [44] is applied as the network structure for generating composition masks $\{M_{ref}^w, M_{tar}^w\}$. Since the image pairs are usually captured at different times, the presence of moving objects can affect the visual appearance of the stitched image, especially in the seam region. As shown in (a) of Fig. 3, taking a human subject

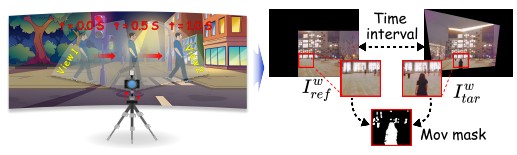

(a) Moving object challenge    (b) Moving mask generation

Figure 3: The generation process of the motion mask during image capturing.

as an example, when capturing images from two viewpoints, the man in the scene may move during the interval, causing him to appear in different positions across the two viewpoints. In order to avoid unsmooth stitching result with ghost caused by above challenges, we further introduce a motion loss $\mathcal{L}_M$ in the training process of MSC to improve the robustness to moving objects, which can be described as follows:

$$\mathcal{L}_M = \omega_1 \left\| M^m I^s - M^m I_{ref}^w \right\|_1 + \omega_2 \left\| M^m I^s - M^m I_{tar}^w \right\|_1, \tag{8}$$

where $\{\omega_1, \omega_2\} = \mathcal{S}(\mathcal{D}(M_{ref}^s \cdot I^s), \mathcal{D}(M_{tar}^s \cdot I^s))$, which controls the composition weight of MSC to the region of moving objects between $I_{tar}^w$ and $I_{ref}^w$. $M^s$ denotes the seam mask, which is introduced in supplementary materials. $\mathcal{S}$ denotes the Softmax operation. $\mathcal{D}$ represents the 2D adjacent difference, which can be illustrated as follows:

$$\mathcal{D}(I) = \sum_{i,j} |I(i, j+1) - I(i,j)| + \sum_{i,j} |I(i+1, j) - I(i,j)|, \tag{9}$$

where $i$ and $j$ represent the location index in x and y axis. $M^m$ denotes the motion mask of the warping images, which can be described as follows:

$$M^m = \mathcal{M}\left(\mathcal{B}\left(|I_{ref}^w - I_{tar}^w|, \phi\right), \kappa\right), \tag{10}$$

where $\mathcal{B}$ denotes the binarization operation with the threshold $\phi$. $\mathcal{M}$ represents the morphology filter. $\kappa$ is the kernel of $\mathcal{M}$ with the size of 3. Therefore, under the constraints of $\mathcal{L}_M$, MSC can adaptively ignore the interference of moving objects appearing in overlapping areas on seamless composition, thereby generating visually pleasing stitched images. We additionally apply boundary loss [10] $\mathcal{L}_B$ and smoothness loss [10] $\mathcal{L}_S$ to make the stitched images smooth and clear. The final composition loss is expressed as follows:

$$\mathcal{L}_{com} = \lambda_4 \mathcal{L}_B + \lambda_5 \mathcal{L}_S + \lambda_6 \mathcal{L}_M. \tag{11}$$

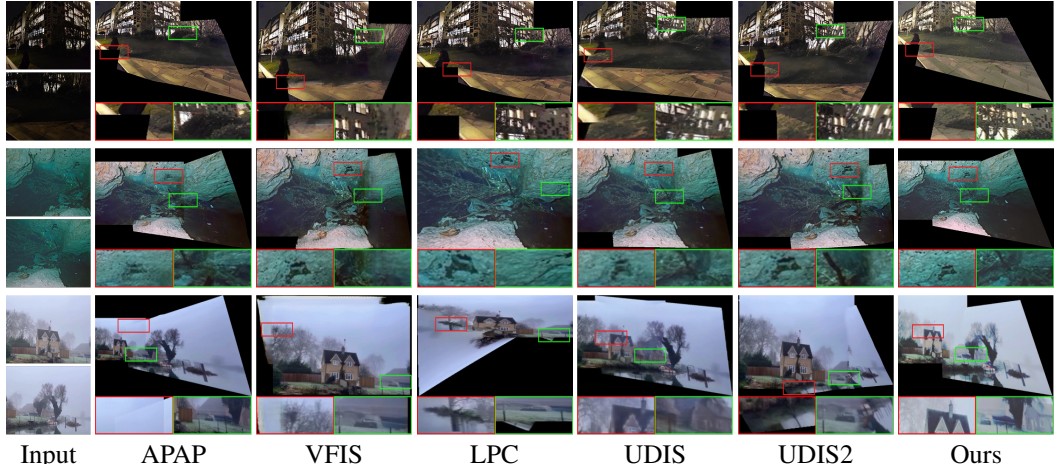

| Input | APAP | VFIS | LPC | UDIS | UDIS2 | Ours |

Figure 5: Visualization results on the ASIS Dataset.

# 4  The Proposed Dataset

In order to comprehensively evaluate image stitching tasks in various harsh environments, we released an Adverse Scene Image Stitching Dataset (ASIS) that integrates in low light, haze and underwater environments. A visual representation of the dataset is shown in (c) of Fig. 1. ASIS contains 2,250 pairs of images, including 750 images each of low-light, underwater, and haze environments, covering 17 scenes such as caves, wrecks and fields. It is worth mentioning that some of the sources of these data come from the Internet and some come from independent photography. The captured images are far from planar structures, which ensures the disparity diversity of the proposed ASIS dataset. More detailed information of ASIS is demonstrated in supplementary materials.

To verify the warping performance of different methods, we provided the homography reference for ASIS. Inspired by [45], the generation process of the reference is illustrated in Fig. 4. We first match image pairs using SIFT [46], however, due to image degradation caused by adverse conditions, it usually suffers from low number and accuracy of matched feature points. Therefore, we manually filter correct matching pairs to prevent the generation of outliers. Moreover, as shown in Fig. 4 (b), where the number of correctly matched pairs is insufficient to generate a reliable homography. Thus, we further select new matching pairs to achieve a more robust homography generation. After obtaining matching pairs, we apply Random sample consensus (RANSAC) [47] to generate the optimal homography and perform preliminary distortion to verify the performance of the reference.

# 5  Experiments

## 5.1  Implement Details

The training process is divided into three steps. First, we pre-train the DIR module with a learning rate of 1e-4 and an epoch of 200. LSRW [48] and UIEBD [49] are used as training datasets for low-light and underwater image enhancement tasks respectively. We also synthesize the training dataset for single image dehazing from VOC [50] according to the atmospheric scattering model [51]. Subsequently, we trained the BCDA module with DIR with an epoch of 160. We then use the above datasets to synthesize the homography training datasets through the synthesis method of VFIS [8]. During the training process, the $\mu$ and $\sigma$ are initialized to 0 and 32 respectively. M, N, $\sigma_{\mathrm{GT}}$, $\phi$ are set to 8, 6, 2 and 20. Batch size and learning rate are set to 16 and 5e-5. Finally, we trained the MSC with a learning rate of 1e-4 and an epoch of 100. $\lambda_1$, $\lambda_2$, $\lambda_3$, $\lambda_4$, $\lambda_5$ and $\lambda_6$ are set to 2, 1, 100, 100, 1 and 1. All the experiments are conducted on PyTorch with NVIDIA RTX 4090 GPU.

Table 1: Quantitative comparison for image stitching under adverse environment. ↑ indicates that higher values correspond to superior outcomes. The top-performing and second-best results are highlighted in red and blue.

| Low-light Environment | | | | | | | | | | | | | | | | |
|---|---|---|---|---|---|---|---|---|---|---|---|---|---|---|---|---|
| Method | Low-light Image | | | | | HBLALS | | | | | NeRCo | | | | | Ours |
| | APAP | VFIS | LPC | UDIS | UDIS2 | APAP | VFIS | LPC | UDIS | UDIS2 | APAP | VFIS | LPC | UDIS | UDIS2 | |
| PSNR↑ | 25.792 | 23.002 | 24.465 | 24.910 | 24.742 | 27.461 | 22.048 | 25.346 | 25.867 | 25.890 | 28.244 | 22.244 | 25.717 | 26.340 | 26.053 | 30.506 |
| SSIM↑ | 0.876 | 0.869 | 0.879 | 0.914 | 0.905 | 0.897 | 0.835 | 0.887 | 0.919 | 0.914 | 0.906 | 0.863 | 0.883 | 0.935 | 0.927 | 0.939 |
| SIQE↑ | 10.352 | 12.329 | 10.861 | 11.372 | 11.493 | 8.786 | 11.897 | 11.062 | 10.197 | 10.437 | 8.507 | 12.578 | 10.837 | 11.089 | 9.688 | 12.854 |
| NIQE↓ | 4.402 | 3.671 | 4.402 | 4.261 | 4.679 | 4.500 | 3.312 | 3.655 | 3.596 | 4.234 | 4.168 | 3.263 | 4.350 | 4.178 | 5.103 | 3.186 |
| Underwater Environment | | | | | | | | | | | | | | | | |
| Method | Underwater Image | | | | | HBLALS | | | | | WaterFlow | | | | | Ours |
| | APAP | VFIS | LPC | UDIS | UDIS2 | APAP | VFIS | LPC | UDIS | UDIS2 | APAP | VFIS | LPC | UDIS | UDIS2 | |
| PSNR↑ | 24.631 | 23.740 | 23.054 | 25.105 | 25.354 | 25.192 | 22.399 | 23.299 | 24.851 | 24.897 | 24.251 | 21.276 | 22.458 | 23.706 | 24.776 | 28.607 |
| SSIM↑ | 0.856 | 0.838 | 0.826 | 0.876 | 0.889 | 0.863 | 0.796 | 0.823 | 0.863 | 0.868 | 0.862 | 0.787 | 0.826 | 0.806 | 0.861 | 0.918 |
| SIQE↑ | 5.657 | 5.668 | 4.012 | 5.646 | 6.115 | 5.456 | 6.089 | 5.313 | 5.901 | 6.152 | 4.934 | 5.998 | 5.919 | 5.086 | 6.057 | 6.444 |
| NIQE↓ | 4.210 | 3.998 | 4.510 | 4.941 | 6.444 | 3.918 | 3.531 | 4.000 | 4.496 | 5.849 | 4.108 | 3.715 | 4.203 | 4.586 | 6.067 | 4.202 |
| Haze Environment | | | | | | | | | | | | | | | | |
| Method | Haze Image | | | | | HBLALS | | | | | WGWS | | | | | Ours |
| | APAP | VFIS | LPC | UDIS | UDIS2 | APAP | VFIS | LPC | UDIS | UDIS2 | APAP | VFIS | LPC | UDIS | UDIS2 | |
| PSNR↑ | 25.065 | 22.037 | 23.657 | 23.532 | 24.690 | 27.615 | 23.969 | 26.135 | 26.009 | 27.065 | 27.530 | 24.038 | 26.380 | 25.836 | 27.245 | 31.136 |
| SSIM↑ | 0.911 | 0.866 | 0.900 | 0.909 | 0.928 | 0.941 | 0.869 | 0.928 | 0.921 | 0.939 | 0.936 | 0.878 | 0.929 | 0.922 | 0.941 | 0.963 |
| SIQE↑ | 7.785 | 6.817 | 6.251 | 6.970 | 7.637 | 6.885 | 7.524 | 6.798 | 7.009 | 7.120 | 7.908 | 8.000 | 7.361 | 7.470 | 7.510 | 8.057 |
| NIQE↓ | 5.632 | 5.297 | 5.681 | 5.629 | 6.224 | 5.255 | 4.972 | 5.311 | 5.434 | 5.990 | 5.687 | 5.160 | 5.747 | 5.547 | 6.190 | 4.970 |

## 5.2 Comparison with Existing Methods

This method mainly studies the stitching algorithm in low-light, haze and underwater environments. For a fair comparison, we first use a unified restoration framework HBLALS [52] and environment-specific restoration methods NeRCo [53], WaterFlow [54] and WGWS [55] respectively to obtain images closer to ideal conditions. We then applied the these clear images to representative stitching algorithms, including traditional APAP[14], LPC [56], and deep learning-based VFIS [8], UDIS [9], and UDIS2 [10]. It is worth mentioning that we also applied the stitching algorithm to the original images without restoration to verify the effectiveness of the restoration method for image stitching.

The first row in Fig. 5 shows the stitching comparison of images restored using HBLALS in low-light scenes. Given the resolution differences across stitching methods, we resize them just for a consistent and aesthetic display. APAP fails to achieve a smooth transition, resulting in obvious seams. VFIS, UDIS and UDIS2 are severely misaligned in the green zoom-in region. LPC breaks the consistency of the road structure in the red zoom-in region. The second row shows the stitching results in the underwater environment, where severe color distortions and similar rock textures impact the stitching performance. LPC causes the loss of distinctive details from the reference image. VFIS and UDIS2 disrupt the reef structures in the overlapping region, resulting in visually unappealing seams. In UDIS, the information within the red box exhibits noticeable blurring, while APAP introduces prominent artifacts in the red box and visible seams in the green box. The third row shows the visualization in the haze environment. Most methods suffer from significant misalignment during the stitching process, resulting in a loss of scene information. Only APAP correctly estimates the image distortion, however introducing significant visual interference at the seam region.

Tab. 1 shows the quantitative results of all methods. For a comprehensive comparison of the proposed methods, we verify the alignment effects by calculating PSNR and SSIM on the overlapping regions. Larger values represent better warping effects. Subsequently, the stitching effect is evaluated through the stitched image quality evaluator (SIQE) [57]. We additionally quoted the Natural Image Quality Evaluator (NIQE)[58] to evaluate the quality of the reconstructed image. A higher SIQE index indicates better stitching effect, and a lower NIQE index indicates better image quality.

It is evident that the proposed method outperforms all others in PSNR and SSIM, validating its effectiveness in warping estimation. Additionally, the SIQE of the proposed method also exceeds other methods in all adverse environments, confirming that our method can greatly reduce the impact of image degradation during stitching. Our NIQE surpasses all methods in low-light and

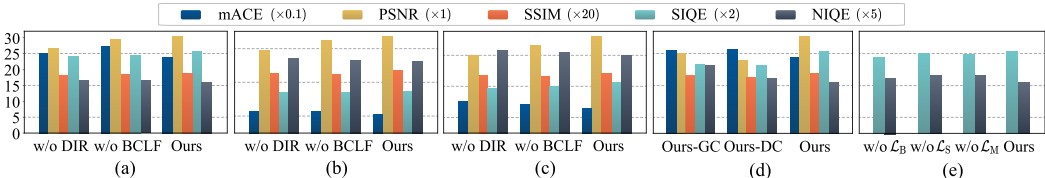

Figure 7: Evaluation on ablation experiment.

haze environments but does not exhibit advantages in underwater environment. This is because underwater scene images are mostly captured in close range with large disparities between image pairs. Therefore, correct warping leads to excessive invalid black regions in the generated images, affecting the evaluation of image quality metrics. Considering all metrics, our method achieves significant advantages in warping and stitching performance under adverse conditions.

## 5.3 Ablation Study

### 5.3.1 Study on Bidirectional-Consistency Learning Framework:

We conducted ablation studies across different environments to verify the effectiveness of the learning framework. Specifically, we first abandoned DIR (w/o DIR) to verify the effect of image restoration on the proposed method. Subsequently, instead of employing BCLF, we only concatenate DIR with homography estimation module (w/o BCLF) to verify the effect of our learning framework. We additionally utilized the mean Average Corner Error (mACE) [59] to validate the performance of the homography estimation. A smaller mACE indicates a more accurate estimation. Fig. 6 illustrates the performance in three adverse environments. It is worth noting that fine

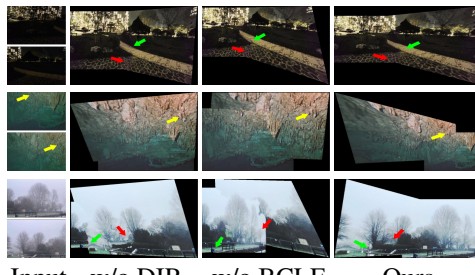

Input   w/o DIR   w/o BCLF   Ours

Figure 6: Visualization results for the study on DIR and BCLF.

details in degradation scenes are difficult to discern by the human eye. Therefore, we additionally restored the results of the unenhanced version in qualitative comparison to facilitate readers in comparing the stitching effects.

In the low-light example, w/o DIR produces artifacts on the floor tiles pointed by the red arrows, destroying the original texture. w/o BCLF suffers from a clear misalignment on the path pointed by the green arrow. In the underwater environment, facing a reef group with a similar structure, neither w/o DIR nor w/o BCLF considers of the correspondence between the input pairs, resulting in a misalignment of the stitching results. In the haze environment, both w/o DIR and w/o BCLF misestimate the deformation, ultimately causing the results to lose a large amount of scene information. (a)-(c) of Fig. 7 depict the quantitative results of the scenarios mentioned above. In summary, the proposed enhancement method demonstrates significant effectiveness under adverse environments.

### 5.3.2 Study on Recursive Parameterized Homography Estimation:

We further conducted ablation studies on RPHE to validate its performance in homography estimation. Specifically, we adopted traditional global correlation (Ours-GC) [9] and dynamic correlation (Ours-DC) [10] as a replacement for the parameterized cost volume. Fig. 8 presents the visual results in low-light environment. The yellow rectangle indicates the overlap ratio of image pairs. The image pairs in the first example face the challenge of a low overlap ratio. Compared with the ours stitching result, both results of Ours-GC and Ours-GC fails on matching the limited scene struc-

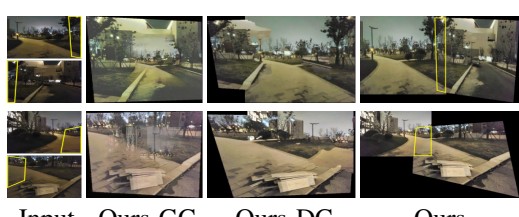

Input   Ours-GC   Ours-DC   Ours

Figure 8: Visualization results for the ablation study on the RPHE.

ture. In the second example, the overlapping region mainly consists of ground areas with fewer fine details, posing significant challenges for stitching. Ours-GC suffers from a significant loss of scene information, while noticeable misalignment occurs in Ours-DC. Only the proposed method achieves accurate stitching results in limited effective areas. Quantitative results of this ablation study are shown in Fig. 7 (d), which also validate the advantages of the proposed homography estimation strategy in all metrics.

### 5.3.3 Study on Motion-Tolerant Seamless Composition:

We ablate $\mathcal{L}_B$, $\mathcal{L}_S$, and $\mathcal{L}_M$ respectively to verify the effect of losses on the image composition stage. The visualization results are shown in Fig. 9. In this example, due to the time difference in image capturing, the character in the red frame has moved from the back of the car to the front of the car. The reconstructed scenes of w/o $\mathcal{L}_B$, w/o $\mathcal{L}_S$ and w/o $\mathcal{L}_M$ all show the same person at different times in the stitched images. At the same time, both w/o $\mathcal{L}_S$ and w/o $\mathcal{L}_M$ suffer from serious information loss in the human leg. Furthermore, as shown in the green frame, obvious seam differences appears in the results of w/o $\mathcal{L}_B$ and w/o $\mathcal{L}_S$. Only the proposed method ensures the temporal uniformity while smoothing the seam difference.

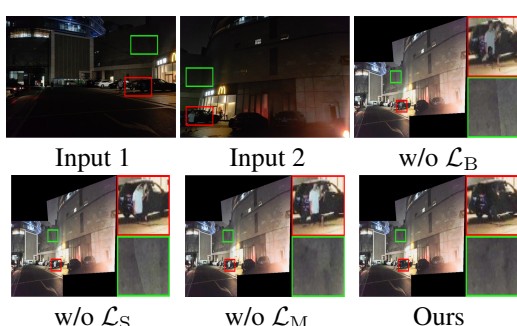

Figure 9: Visualization results for the ablation study on the MSC.

(e) of Fig. 7 shows the qualitative metrics of the reconstruction module. Since all experiments are performed under the same image warping, we only use image quality metrics for evaluation. Both qualitative and quantitative experiments demonstrate the effectiveness of the proposed constraints for reconstructing the network.

## 6 Limitations

The proposed motion constraint only improves stitching performance for small-range movements in dynamic scenes. For instance, when a pedestrian moves within the overlapping region of two images, our method effectively suppresses visual ghosting. However, if the movement extends beyond the overlap, the network cannot recognize it as the same object, leading to duplicate appearances in the final stitched image. This limitation arises because the motion loss focuses solely on movements within the overlapping area and fails to handle large displacements. Addressing fast, large-scale motions while ensuring non-redundant targets in the output remains challenging and may require integrating explicit object detection with a generative model to reconstruct occluded backgrounds around duplicated targets.

## 7 Conclusion

We propose an adverse condition-tolerant image stitching network. It features a bidirectional consistency learning framework, which iteratively optimizes differentiable restoration and Gaussian-encoded homography estimation for reliable alignment. Additionally, motion constraint is proposed in the seamless composition stage, suppressing artifacts from moving objects. We also construct the first adverse scene image stitching dataset, covering diverse low-light, haze, and underwater scenarios. Extensive experiments on the proposed dataset demonstrate the stitching performance of our method under adverse conditions.

## Acknowledgments

This work is partially supported by the China Postdoctoral Science Foundation (No. 2023M730741)); in part by the National Natural Science Foundation of China (Nos. 62302078, 62372080); and in part by the Fundamental Research Funds for the Central Universities (No. 3132025276).

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

# A    Technical Appendices and Supplementary Material

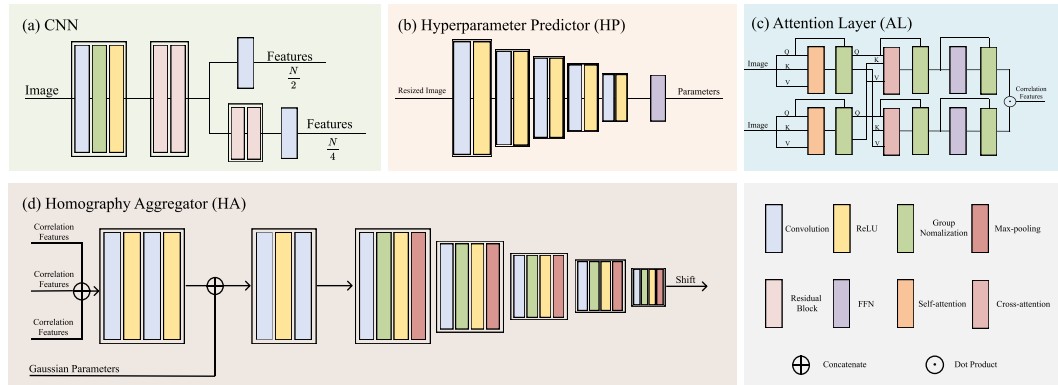

Figure 10: Detailed structure of CNN (a), Hyperparameter Predictor (b), Attention Layer (c) and Homography Aggregator (HA).

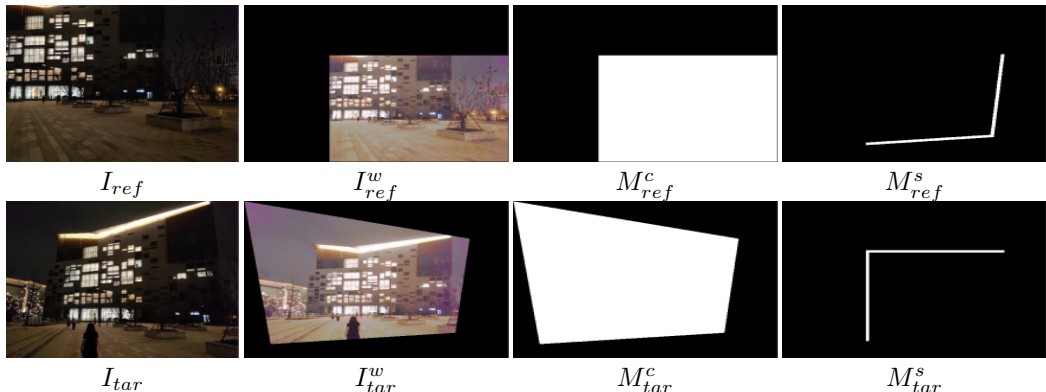

Figure 11: Visualization of the seam mask.

## A.1    Detailed Network Architecture

Fig. 10 demonstrates the detailed network structure of the proposed Pyramid-CNN (a), Hyperparameter Predictor (b), Attention Layer (c) and Homography Aggregator (d).

## A.2    Details in MSC

The generation process of the introduced seam mask $\{M_{ref}^s, M_{tar}^s\}$ can be formulated as follows:

$$
\begin{aligned}
\nabla M_{ref}^c &= \left|M_{ref,i,j}^c - M_{ref,i-1,j}^c\right| + \left|M_{ref,i,j}^c - M_{ref,i,j-1}^c\right|, \\
\nabla M_{tar}^c &= \left|M_{tar,i,j}^c - M_{tar,i-1,j}^c\right| + \left|M_{tar,i,j}^c - M_{tar,i,j-1}^c\right|,
\end{aligned}
\tag{12}
$$

$$
\begin{aligned}
M_{ref}^s &= \mathcal{C}(\mathcal{E}(\mathcal{E}(\mathcal{E}(\mathcal{E}(\nabla M_{tar}^c))))) \odot M_{ref}^c, \\
M_{tar}^s &= \mathcal{C}(\mathcal{E}(\mathcal{E}(\mathcal{E}(\mathcal{E}(\nabla M_{ref}^c))))) \odot M_{tar}^c,
\end{aligned}
\tag{13}
$$

where $i, j$ are index of the Cartesian coordinates. $\{M_{ref}^c, M_{tar}^c\}$ are the content mask, which replace the original pixel of image into all-in-one matrix. $\mathcal{E}$ is the convolution layer with 3×3 SOBEL filters. $\mathcal{C}$ clip image to 0-1. The visualization of the $\{M_{ref}^c, M_{tar}^c, M_{ref}^s, M_{tar}^s\}$ is also shown in Fig. 11.

## A.3    More Demonstration on ASIS Dataset

We additionally provide examples of the dataset in different scenarios under adverse environments, which is demonsterated in Fig. 12. It is worth mentioning that the low-light scene image pairs are

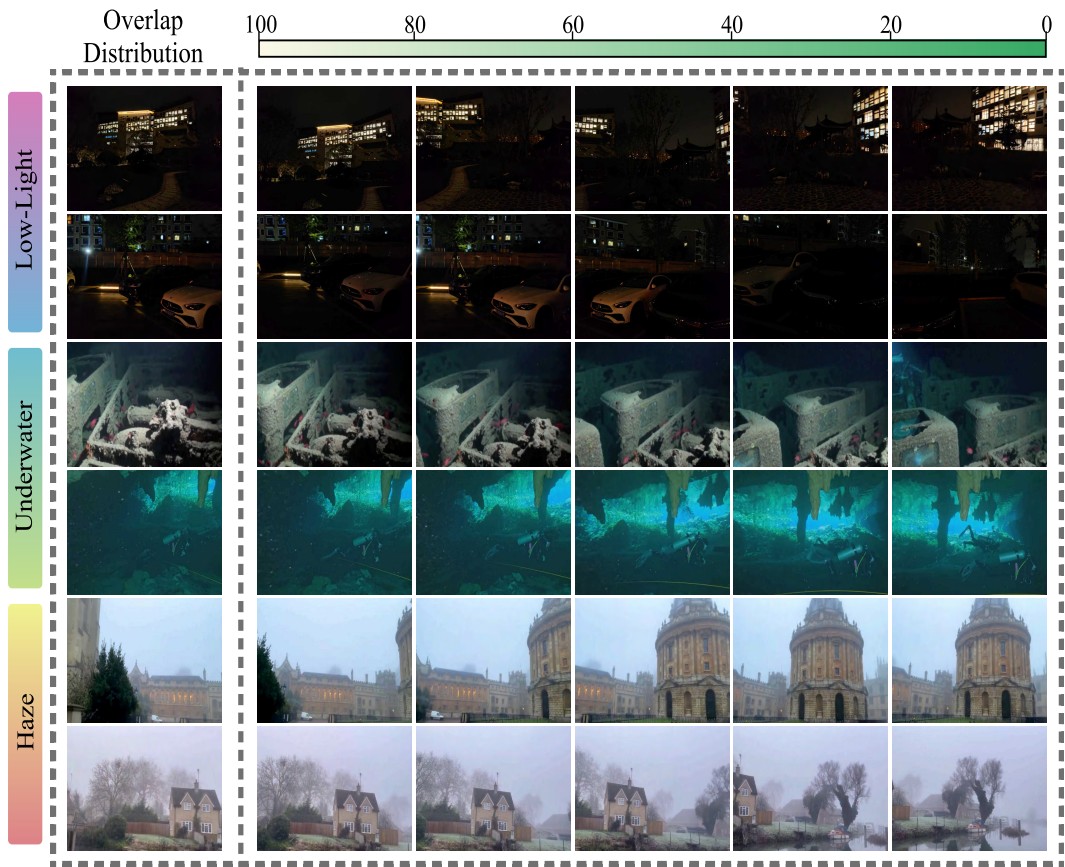

Figure 12: Examples of image pairs under three adverse environment in order of baseline from small to large.

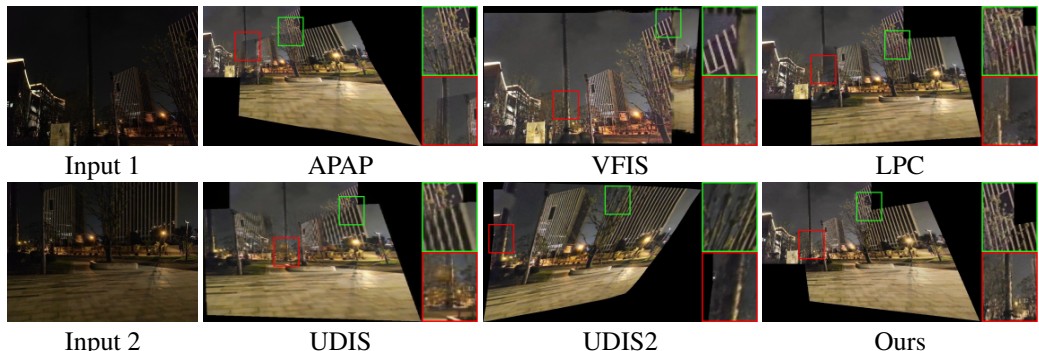

Figure 13: Visualization results on the low-light environment.

collected manually, and the image pairs of the underwater and haze environment are collected from the Internet [2]

---

[2] https://www.youtube.com/watch?v=93bWdgI69To
https://www.youtube.com/watch?v=dBMrRJWrFEU
https://www.youtube.com/watch?v=G5Mr3NuHjaI
https://www.youtube.com/watch?v=D1KsEOUqCEU
https://www.youtube.com/watch?v=yCoNJHqWYnU&list=RDyCoNJHqWYnU&start_radio=1&t=73s

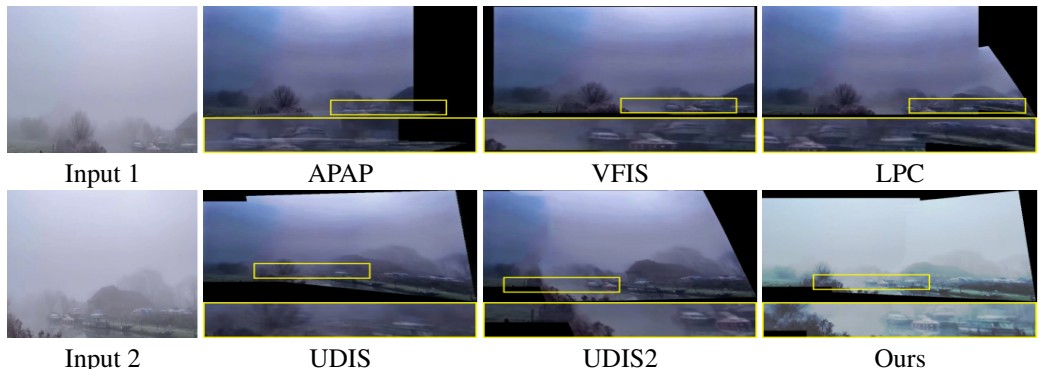

| | | | |
|---|---|---|---|
| Input 1 | APAP | VFIS | LPC |
| Input 2 | UDIS | UDIS2 | Ours |

Figure 14: Visualization results on the haze environment.

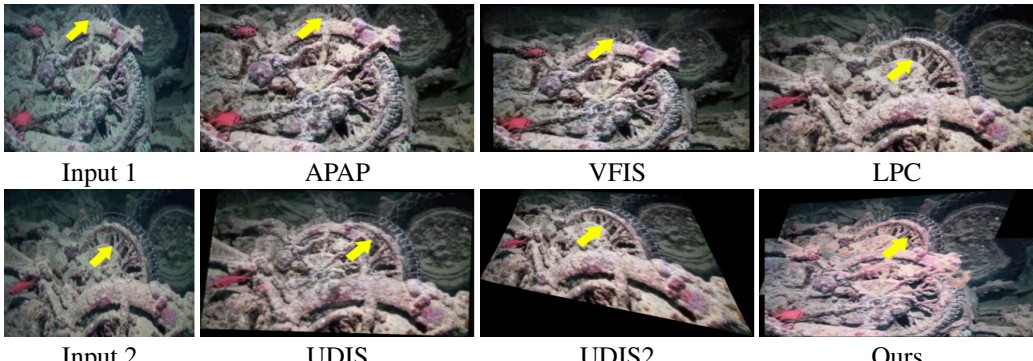

| | | | |
|---|---|---|---|
| Input 1 | APAP | VFIS | LPC |
| Input 2 | UDIS | UDIS2 | Ours |

Figure 15: Visualization results on the underwater environment.

## A.4 More Visual Comparisons

We additionally provide examples for evaluating the performance in low-light,haze and underwater environment. We use the unified restortation framework HBALALS [52] combined with APAP [14], VFIS [8], LPC [56], UDIS [9] and UDIS2 [10] to compare with our method. The qualitative results are shown in Fig. 13, Fig. 14 and Fig. 15.

