# OpenReview forum: "Image Stitching in Adverse Condition: A Bidirectional-Consistency Learning Framework and Benchmark"
_NeurIPS.cc/2025/Conference — NeurIPS 2025 poster_

### Official Review · Reviewer_o2QN · 2025-06-25

**Clarity:** 3
**Significance:** 3
**Originality:** 2
**Rating:** 4
**Confidence:** 3

**Summary:**

The authors propose an image stitching method designed for adverse conditions by leveraging a Gaussian distribution to encode the displacement space.
To address image degradation caused by severe weather, they introduce a differentiable image restoration module, whose output is subsequently integrated into MSC.
Although image stitching has been studied under various conditions, the authors construct a dedicated dataset for extreme scenarios with insufficient visual cues, providing adverse weather image pairs along with corresponding ground truth for reliable evaluation.

**Questions:**

1. The authors should include a more thorough comparison with existing panoramic datasets, such as the UDIS dataset.

2. The authors should analyze the proposed model’s performance under normal (non-adverse) conditions.

**Ethical Concerns:**

["NO or VERY MINOR ethics concerns only"]

**Final Justification:**

By reading the rebuttal, the concerns raised from the paper have been resolved. The fact that the proposed method performs well not only under adverse conditions but also in general normal scenarios suggests that it is a well-designed approach. Therefore, I am keeping my score unchanged.

**Limitations:**

There is a lack of discussion on limitations or future work. To ensure the proposed model is applicable in a wide range of scenarios, it should be evaluated not only under extreme conditions such as adverse weather but also under normal conditions. However, such analysis is missing, making it difficult to identify the model’s limitations in general settings. The authors should address this gap by including a discussion of relevant limitations, which could help pave the way for future improvements.

**Paper Formatting Concerns:**

no issue i found

**Quality:**

3

**Strengths And Weaknesses:**

Strengths
1. The paper presents a reliable method for image stitching in degraded conditions. The authors effectively integrate the image restoration and warping modules in an efficient and cohesive manner.
2. A new dataset including images from adverse weather situations and their ground truth.

Weaknesses
1. Insufficient comparison with existing panoramic datasets, such as the UDIS dataset.
2. Lack of analysis of the proposed model’s performance under normal (non-adverse) conditions.

---

> ### Author Rebuttal · Authors · 2025-07-30
>
> Thank you for your valuable feedback. I have carefully considered your suggestions and concerns, and will revise the manuscript accordingly. Please find my detailed responses to your comments below:
>
> ---
> **Answer for Q1&Q2**:
> Thank you for your suggestion. Our method can also serve as a general-purpose image stitching method under normal conditions. To demonstrate this, we conducted additional comparison experiments on the standard UDIS dataset. As shown in Table 1, the proposed method is quantitatively compared with other stitching methods on the UDIS dataset.
>
> It is worth noting that since the UDIS dataset features normal lighting conditions without rain or fog, it differs from the adverse environments discussed in the main paper. Therefore, unlike the comparisons in the main text that involve combining task-specific restoration methods with stitching algorithms, here we directly compare pure stitching methods.
>
> To ensure a fair comparison, and in contrast to the experimental protocol described in the manuscript (which uses synthetically generated pairs from adverse-environment datasets), we re-trained our method on the UDIS dataset using the Adam optimizer. The training was conducted for 100 epochs with a learning rate of 1e-4, and the retrained model was used for testing to validate the effectiveness of the proposed method under normal conditions.
>
> Due to NIPS rebuttal policy restrictions, we are unable to provide visual comparisons. However, as shown in the table, our method still achieves a slight advantage under normal conditions, with a 1% improvement in PSNR, 2% in SSIM, 2% in NIQE, and 12% in SIQE compared to the best-performing existing methods. This demonstrates the generalizability of our proposed approach.
>
> Table 1: Quantitative comparison for image stitching under normal environment (UDIS dataset).
>
> ---
> | Metric | APAP   | VFIS   | LPC    | UDIS     | UDIS2  | Ours       |      | APAP   | VFIS   | LPC    | UDIS  | UDIS2    | Ours       |
> | ------ | ------ | ------ | ------ | -------- | ------ | ---------- | ---- | ------ | ------ | ------ | ------ | -------- | ---------- |
> | PSNR  ($\uparrow$)   | 23.79 | 20.04 | 22.59 | 21.17  | 25.43 | **25.69** | NIQE($\downarrow$) | 4.95  | 4.25  | 4.47  | 4.31    | 4.45  | **4.16**  |
> | SSIM ($\uparrow$)    | 0.79  | 0.58  | 0.73  | 0.64 | 0.83  | **0.85**  | SIQE ($\uparrow$) | 14.80 | 24.00 | 14.47  | 30.29| 20.44 | **33.93** |
> ---
>
> ---
> **Answer for Limitation**:
> Thank you for your suggestion. Regarding the applicability of our model in normal conditions, please refer to our response to Q1 and Q2. The quantitative results in Table 1 demonstrate the effectiveness of the proposed architecture for image stitching under normal conditions.

---

> > ### Comment · Reviewer_o2QN · 2025-08-05
> >
> > Thank you for the rebuttal in response to my questions. The authors’ clarifications have addressed most of the concerns I raised. The authors stated, "Therefore, unlike the comparisons in the main text that involve combining task-specific restoration methods with stitching algorithms, here we directly compare pure stitching methods." In that case, was the same condition applied to the authors' proposed method as well — specifically, was BCDA removed for a fair comparison in this setup?

---

> > > ### Author Response · Authors · 2025-08-05
> > >
> > > Thank you for your feedback. When conducting comparisons on UDIS dataset, we ensure that the proposed method is evaluated under the same conditions to guarantee fair and consistent benchmarking. Specifically, during the homography estimation stage, the restoration-related BCDA algorithm is removed from the proposed pipeline, leaving only the parameterized iterative estimation component. This adjustment allows for a more direct and consistent evaluation of the proposed method's stitching performance under normal conditions.

---

> > > > ### Comment · Reviewer_o2QN · 2025-08-05
> > > >
> > > > Thank you for the detailed response. The concerns raised from the paper have been resolved, and I will keep my current rating.

---

### Official Review · Reviewer_kt8k · 2025-07-01

**Clarity:** 3
**Significance:** 2
**Originality:** 3
**Rating:** 4
**Confidence:** 3

**Summary:**

The article proposes a robust deep image stitching network called ACDIS, which addresses the impact of adverse conditions such as complex weather, lighting changes, and dynamic scene motions in real - world scenarios on image stitching. It introduces a bidirectional consistency learning framework that combines differentiable image restoration with Gaussian - distributed homography estimation, achieving reliable alignment.

**Questions:**

Please see the weaknesses.

**Ethical Concerns:**

["NO or VERY MINOR ethics concerns only"]

**Final Justification:**

The author solved my problem and I raised the score to 4.

**Limitations:**

yes

**Quality:**

2

**Strengths And Weaknesses:**

Strengths：
1. The authors constructed the first image stitching dataset named ASIS, which includes various adverse scenarios such as low - light conditions, foggy weather, and underwater environments. They conducted extensive experiments to verify the effectiveness of the proposed method.
2. The proposed ASIS dataset provides a valuable benchmark for the evaluation of image stitching tasks under adverse conditions

Weaknesses：
1. Although the article introduces motion constraints in the seamless stitching network, the method may only be able to handle dynamic scene changes within a certain range. For more complex dynamic scenes, such as those with multiple moving objects or objects moving at high speeds, the article does not elaborate on how to more effectively address the image stitching problem under these complex dynamic scenarios. Further exploration and improvement of the motion constraint mechanism could be made to adapt to a wider range of dynamic scenes.
2. The lack of interpretability is a notable drawback. Although the paper provides detailed technical descriptions, it falls short in offering in-depth explanations and analyses of the principles and processes of the core methods, such as the bidirectional-consistency learning framework and the recursive parameterized homography estimation module. For instance, the paper does not provide sufficient intuitive explanations regarding why the bidirectional optimization approach can enhance alignment reliability more effectively, as well as the underlying mechanisms of how the recursive parameterized homography estimation module gradually approximates the true homography transformation during the iterative process.
3. The proposed ACDIS network, which includes multiple modules and iterative optimization processes, may suffer from relatively low computational efficiency. This could be a significant limitation in practical applications, especially in scenarios where real-time performance is required. The authors should consider optimizing the network structure or algorithmic flow to improve its speed and efficiency.

---

> ### Author Rebuttal · Authors · 2025-07-30
>
> Thank you for your valuable comments. In response to your suggestions and concerns, I will provide detailed explanations and revise the manuscript accordingly. Please find my responses below.
>
> ---
> **Ans for W1**: Thank you for your feedback. The proposed motion constraint currently improves stitching performance in dynamic scenes to a certain extent. Taking pedestrians in road scenarios as an example: when a person moves during the acquisition of the image pair, our method can effectively suppress visual ghosting in the stitched result—but only if the motion remains within the overlapping region of the two images. If the pedestrian's movement exceeds this overlapping area, the network cannot recognize them as the same target, resulting in duplicated appearances in the final stitched image. This is because the core of our motion loss lies in capturing moving content within the overlap, and it cannot handle motion beyond that scope.
>
> Addressing high-speed motion with large displacements (beyond the overlapping area), while ensuring that only non-redundant targets remain in the final output, remains highly challenging. This may require integrating explicit object detection modules with generative models to inpaint the occluded background around repeated targets using contextual information. We will include a discussion of this limitation in the Limitations section of the manuscript.
>
> ---
> **W2(1)**: “Why the bidirectional optimization approach can enhance alignment reliability more effectively”
>
> **Ans**:
> The proposed optimization framework is proposed to address the issue of inconsistent restoration. In image alignment tasks under adverse conditions, we aim for the backbone network to produce restoration results that are equivariant under spatial transformations. This objective can be formally expressed as:
>
> $$\Phi\left(\mathcal{W}\left(I_{\text{tar}} ; \mathbf{H}\right)\right) = \mathcal{W}\left(\Phi\left(I_{\text{tar}}\right); \mathbf{H}\right),$$
>
> where $\mathcal{W}$ denotes the warping operation based on homography $\mathbf{H}$, $I_{\text{tar}}$ is the target image, and $\Phi$ is the restoration module.
>
> In this ideal case, the restored features from corresponding positions would perfectly match. However, in practice, it is difficult for restoration networks to maintain equivariance under homography transformations, which involve translation, rotation, scaling, shearing, aspect ratio change, and perspective distortion. This leads to inconsistent features between corresponding points after enhancement \[1], ultimately degrading the performance of homography estimation—an observation consistent with the results shown in Fig. 1(a) of the manuscript.
>
> To address this issue, we depart from the conventional sequential paradigm of restoration followed by warping. Instead, we propose an alternating scheme within our recursive homography estimation framework, referred to as *restoration-guided warping*. As illustrated in Fig. 1(b), the image warping module and restoration module are interleaved during the iterative process, which can be formulated as:
>
> $$E_{\text{tar}}^n = \Phi\left(\mathcal{W}\left(I_{\text{tar}} ; \hat{\mathbf{H}}^n\right)\right), \quad
> \hat{\mathbf{H}}^{n+1} = \Psi\left(I_{\text{ref}} ; E_{\text{tar}}^n\right),$$
>
> where $\Psi$ denotes the homography estimation module, and $E_{\text{tar}}^n$ and $\mathbf{H}^n$ represent the enhanced warped image and the estimated homography at the $n$-th iteration, respectively.
>
> As the iteration proceeds, the warped image $I_{\text{tar}}^w$ becomes increasingly aligned with the reference image $I_{\text{ref}}$, allowing the DIR module to produce more consistent restoration in the overlapping regions, which in turn improves the accuracy of homography estimation. The effectiveness of this mechanism is demonstrated in Fig. 1(b) of the main text. It is also important to note that although the framework performs multiple iterations, the restoration network (DIR) shares weights across iterations, and thus does not incur any additional parameter overhead.
>
> [1] Gift: Learning transformation-invariant dense visual descriptors via group CNNs.
>
> ---
> **W2(2)**: “How the homography estimation module gradually approximates the true homography transformation during the iterative process.”
>
> **Ans**: The iterative process in our homography estimation network can be summarized in three main steps to better clarify the mechanism:
>
> 1. In each iteration, disparity candidates are sampled using the current mean $\mu$ and standard deviation $\sigma$. Specifically, at the $i$-th iteration, candidates are uniformly sampled within the interval $[ \mu_i - \sigma_i, \mu_i + \sigma_i ]$. For the initial iteration, $\mu_0$ is initialized to 0 and $\sigma_0$ is set to 32 (as mentioned in the experimental details of the manuscript).
>
> 2. The matching cost is computed based on the sampled disparities of the current iteration, which is then used to predict the update step $\Delta_i$.
>
> 3. According to Equation (3) in the manuscript, the predicted step $\Delta_i$ is used to update the current $\mu$ and $\sigma$ according to follows formulation:
>
> $$\qquad\qquad\sigma_i^{t}  =\sigma_i^{t-1}-\frac{1}{2}\left(\frac{(\sigma_i^{t-1})^2-\sigma_G^2-(\Delta^{t-1})^2}{ (\sigma_i^{t-1})^3}-\frac{1}{\sigma_i^{t-1}}+\frac{\sigma_i^{t-1}}{\sigma_G^2}\right),
> 		\mu_i^{t}  =\mu_i^{t-1}-\left(-\frac{\Delta^{t-1}}{2}\left(\frac{1}{(\sigma_i^{t-1})^2}+\frac{1}{\sigma_G^2}\right)\right).$$
>  &nbsp; &nbsp; &nbsp; where $\sigma_G$ is a predefined parameter.
>
> After $N$ iterations, the final updated $\mu_N$ is taken as the predicted displacement. A direct linear transformation is then applied to obtain the final homography matrix. Meanwhile, $\sigma$ progressively decreases throughout the iterations, making the sampling more concentrated and thus improving the accuracy and stability of the final prediction.
>
> The advantage of this approach lies in its ability to maintain differentiability throughout the sampling and matching process during training. Moreover, by introducing JS divergence, the distribution gradually converges toward a standard Gaussian, which further accelerates model convergence and enhances generalization. The optimization process can be formulated as:
> $$\mathcal{J}_i=\frac{1}{2}\left(F\left(\mathcal{N}_G \| \mathcal{N}_i\right)+F\left( \mathcal{N}_i \| \mathcal{N}_G\right)\right),$$
>
> where $\mathcal{N}_i$ is the short form of $\mathcal{N}\left(\mu_i, \sigma_i^2\right)$. $\mathcal{N}_G=\mathcal{N}\left(\mu_G, \sigma^2_G\right)$ denotes predefined Gaussian distribution, where $\mu_G$ represents the ground truth offset. $F\left( \cdot  \|\cdot\right)$ represents KL divergence. This combination of random modeling and differentiable optimization effectively addresses the limitations of traditional methods.
>
> ---
> **Ans for W3**: Thank you for your suggestion. Multi-iteration optimization is a widely adopted strategy in current deformation estimation networks. Since image stitching tasks often involve handling large-baseline scenarios, a coarse-to-fine iterative deformation process is crucial for achieving satisfactory stitching performance[1-2]. Although single-iteration strategies can indeed improve runtime, they often lead to a significant drop in final stitching quality.
>
> To ensure practical applicability, our method is designed to strike a balance between stitching accuracy and computational efficiency, aiming to meet real-time requirements. During the design phase, we introduced the following optimizations:
>
> *1. Convergence Acceleration via Gaussian-Encoded Optimization:*
> We ensure the differentiability of the sampling and matching process during training. By introducing a JS divergence constraint, the distribution gradually converges to a standard Gaussian, which accelerates model convergence and improves generalization. This combination of probabilistic modeling and differentiable optimization effectively addresses the limitations of traditional methods.
>
> *2. Lightweight Image Restoration Module:*
> The module is designed to be lightweight, consisting of only 5 convolutional layers. It is solely responsible for predicting the hyperparameters used in restoration. With a computational cost of just 0.07 Gflops.
>
> To further validate the efficiency, we conducted runtime comparison experiments under identical settings.
> Tables below show the processing time per image pair for all methods. The average runtime per pair for our method is 0.336s (low-light), 0.322s (underwater), and 0.344s (foggy). Across all three scenes, the proposed method performs competitively, with runtime only slightly higher than VFIS and its enhanced variant (restoration + VFIS). In future work, we will explore further simplifications of the module architecture and investigate low-iteration optimization strategies.
>
> ||||LLI|||||HBLALS|||||NeRCo||||
> |-|-|-|-|-|-|-|-|-|-|-|-|-|-|-|-|-|
> || APAP|VFIS|LPC|UDIS|UDIS2| APAP|VFIS|&nbsp; LPC|UDIS|UDIS2|APAP|&nbsp; VFIS|&nbsp; LPC|UDIS|UDIS2|Ours|
> |time(s)|2.282|**0.283** | 1.550 | 0.692|0.351|2.313|*0.302*|&nbsp; 1.577|0.710|0.380|2.291|&nbsp; 0.299|&nbsp; 1.562|0.704|0.360|0.336|
> ---
>
> ||||UI|||||HBLALS|||||WFlow||||
> |-|-|-|-|-|-|-|-|-|-|-|-|-|-|-|-|-|
> || APAP|VFIS|LPC|UDIS|UDIS2|APAP|VFIS|&nbsp; LPC|UDIS|UDIS2|APAP|&nbsp; VFIS|&nbsp; LPC|UDIS|UDIS2|Ours|
> |time(s)|2.262|**0.304**|1.602|0.699|0.350|2.274|*0.313*|&nbsp;  1.623|0.710|0.366|2.382|&nbsp; 0.400|&nbsp;  1.695|0.804|0.362|0.322|
> ---
>
> ||||SI|||||HBLALS|||||WGWS||||
> |-|-|-|-|-|-|-|-|-|-|-|-|-|-|-|-|-|
> || APAP|VFIS|LPC|UDIS|UDIS2|APAP|VFIS|&nbsp; LPC|UDIS|UDIS2|APAP|&nbsp; VFIS|&nbsp; LPC|UDIS|UDIS2|Ours|
> |time(s)|2.174|**0.262**|1.499|0.660|0.295|2.180|*0.273*|&nbsp; 0.520|0.679|0.348|2.262|&nbsp; 0.331|&nbsp; 1.573|0.759|0.390|0.334|
> ---
>
>
> [1] Learning edge-preserved image stitching from multi-scale deep homography[J]. Neurocomputing, 2022.
>
> [2] Unsupervised deep image stitching: Reconstructing stitched features to images[J]. IEEE TIP, 2021.

---

> ### Comment · Area_Chair_JwCs · 2025-08-07
>
> Dear Reviewer kt8k,
>
> Please engage in the discussion with the authors and other reviewers as soon as possible.
>
> Thank you.
>
> Best,
>
> AC

---

### Official Review · Reviewer_PgFv · 2025-07-02

**Clarity:** 2
**Significance:** 3
**Originality:** 3
**Rating:** 5
**Confidence:** 4

**Summary:**

This paper proposes a method that effectively handles various adverse factors in image stitching. It presents a novel approach for computing the correlation volume and successfully integrates an image enhancement module with the homography estimation module. As a result, the method demonstrates superior stitching performance. Additionally, the paper introduces a new dataset that reflects these diverse and challenging conditions.

**Questions:**

**Q1.** I have a question regarding the process of generating homographies for training. To learn the Gaussian distribution, it seems both the mean (μ) and standard deviation (σ) must be specified. Could you clarify how these are determined? Specifically, is the mean (μ) sampled from a uniform distribution, and is the final displacement value then sampled from the resulting normal distribution, N(μ, σ)?

**Q2.** Regarding the DIR module, would using a generic CNN be less effective than the current design with its specialized, hand-crafted blocks? Was this alternative considered?

**Q3.** Could you clarify why the mACE metric is not reported in Table 1?

**Q4.** How does the total training time of the proposed method compare to that of other existing methods?

**Ethical Concerns:**

["NO or VERY MINOR ethics concerns only"]

**Final Justification:**

I believe the following strengths of this paper will have a impact on future research:

- It addresses the overconfidence problem by utilizing stochastic predictions rather than conventional deterministic ones.

- It introduces a scalable DIR module, moving beyond naive CNN architectures and integrating it into the stitching learning process.

- It presents a novel dataset for the community.

Therefore, I would like to raise my rating.

**Limitations:**

**L1.** The method requires input images to be of a fixed size.

**L2.** Other limitations are as detailed in the main 'Weaknesses' section.

**Paper Formatting Concerns:**

Minor typos.

**Quality:**

3

**Strengths And Weaknesses:**

**S1.** The effective integration of image enhancement into the homography estimation process enables more accurate results compared to using enhancement merely as a preprocessing step.

**S2.** It proposes a method for utilizing a correlation volume to handle image pairs with large baselines.

**S3.** Factors affecting image quality are incorporated into the DIR module by leveraging inductive biases.

**S4.** The paper contributes a novel benchmark dataset.

**W1.** The DIR module may struggle to address types of image degradation that are not explicitly covered by its pre-defined blocks.

**W2.** The inclusion of a sampling process reduces the method's efficiency.

**W3.** The DIR, BCDA, and MSC modules are trained separately, rather than through a single, end-to-end process.

**W4.** Comparative experiments against other methods on the newly proposed dataset are not included in the paper.

---

> ### Author Rebuttal · Authors · 2025-07-31
>
> Thank you for your valuable comments. I will provide a detailed explanation based on your suggestions and concerns, and revise the manuscript accordingly. Below are my responses to your questions:
>
> ----
> **Ans for W1**:
> Thank you for your valuable feedback. Explicitly designing submodules for specific degradation types does indeed limit adaptability when dealing with unknown or compound degradations.
>
> However, we have paid particular attention to modularity and scalability in the design of the DIR module. Each degradation-specific sub-module is implemented in a plug-and-play manner, allowing flexible extension or replacement to accommodate new degradation types without retraining the entire model. Moreover, all DIP modules are connected to a shared encoder that captures global contextual information. Even when a specific degradation type is not explicitly modeled, the shared features can still compensate for potential performance loss through semantic interaction.
>
> ---
> **Ans for W2**:
> Thank you for your suggestion. The core purpose of adopting uniform sampling is to construct a differentiable and robust optimization module that can model the uncertainty of the current offset distribution, thereby enhancing the model’s robustness to local minima and ambiguous regions.
>
> To further evaluate the efficiency, we conducted a runtime comparison under the same experimental settings.
> The table below lists the processing time per image pair for all methods across low-light, underwater, and foggy scenes.
> It can be observed that our proposed method still performs competitively under sampling conditions, with runtime only slightly higher than VFIS and its enhanced variant (Restoration + VFIS). In future work, we will explore further simplifications of the module architecture and investigate low-iteration optimization strategies.
>
> ||||LLI|||||HBLALS|||||NeRCo||||
> |-|-|-|-|-|-|-|-|-|-|-|-|-|-|-|-|-|
> || APAP|VFIS|&nbsp; LPC|UDIS|UDIS2|&nbsp; APAP|&nbsp; VFIS|&nbsp; LPC|UDIS|UDIS2|&nbsp; APAP|&nbsp; VFIS|&nbsp; LPC|UDIS|UDIS2|Ours|
> |time(s)|2.282|**0.283** | &nbsp; 1.550 | 0.692|0.351|&nbsp;  2.313|&nbsp;  *0.302*|&nbsp; 1.577|0.710|0.380|&nbsp;  2.291|&nbsp;  0.299|&nbsp; 1.562|0.704|0.360|0.336|
> ---
>
> ||||UI|||||HBLALS|||||WFlow||||
> |-|-|-|-|-|-|-|-|-|-|-|-|-|-|-|-|-|
> || APAP|VFIS|&nbsp; LPC|UDIS|UDIS2|&nbsp; APAP|&nbsp; VFIS|&nbsp; LPC|UDIS|UDIS2|&nbsp; APAP|&nbsp; VFIS|&nbsp; LPC|UDIS|UDIS2|Ours|
> |time(s)|2.262|**0.304**|&nbsp;  1.602|0.699|0.350|&nbsp;  2.274|&nbsp;  *0.313*|&nbsp;  1.623|0.710|0.366|&nbsp;  2.382|&nbsp;  0.400|&nbsp;  1.695|0.804|0.362|0.322|
> ---
>
> ||||SI|||||HBLALS|||||WGWS||||
> |-|-|-|-|-|-|-|-|-|-|-|-|-|-|-|-|-|
> || APAP|VFIS|&nbsp; LPC|UDIS|UDIS2|&nbsp; APAP|&nbsp; VFIS|&nbsp; LPC|UDIS|UDIS2|&nbsp; APAP|&nbsp; VFIS|&nbsp; LPC|UDIS|UDIS2|Ours|
> |time(s)|2.174|**0.262**|&nbsp; 1.499|0.660|0.295|&nbsp; 2.180|&nbsp; *0.273*|&nbsp; 0.520|0.679|0.348|&nbsp; 2.262|&nbsp; 0.331|&nbsp; 1.573|0.759|0.390|0.334|
> ---
>
> **Ans for W3** :
> Thank you for your question. First, BCDA and MSC are treated as two independent stages in the image stitching pipeline and are trained separately. The BCDA module focuses on estimating transformation parameters, typically supervised by geometric errors or positional discrepancies; in contrast, the composition stage emphasizes image blending quality and relies on pixel-level or perceptual losses. Due to the significant differences in their numerical scales, gradient directions, and convergence behaviors, end-to-end training cannot effectively optimize both tasks simultaneously. Therefore, similar to other learning-based stitching methods \[1–2], we adopt a staged training strategy.
> For training BCDA and DIR, directly training BCDA disrupts DIR’s pixel-level supervision, reducing its enhancement. To avoid this, we pre-trained DIR for 200 epochs before joint training.
>
> [1] Unsupervised deep image stitching: Reconstructing stitched features to images. TIP, 2021.
> [2] Parallax-tolerant unsupervised deep image stitching. ICCV, 2023.
>
> ---
> **Ans for W4**::
> Thank you for your suggestion. Tab. 1 and Fig. 5 in the manuscript present the quantitative and qualitative comparison of the newly proposed dataset, covering image stitching performance under three challenging conditions.
>
> Besides directly stitching degraded images for baseline comparison, we also restore them using task-specific methods before stitching to ensure fairer evaluation. The results highlight that our method is not just a simple combination, but a tailored solution for stitching under degradation.
>
> ---
> **Ans for Q1**::
> Thank you for your question. The iterative process in our homography estimation network can be summarized in three main steps to better clarify the mechanism:
>
> 1. In each iteration, disparity candidates are sampled using the current mean $\mu$ and standard deviation $\sigma$. Specifically, at the $i$-th iteration, candidates are uniformly sampled within the interval $[ \mu_i - \sigma_i, \mu_i + \sigma_i ]$. For the initial iteration, $\mu_0$ is initialized to 0 and $\sigma_0$ is set to 32 (as mentioned in the experimental details of the manuscript).
>
> 2. The matching cost is computed based on the sampled disparities of the current iteration, which is then used to predict the update step $\Delta_i$.
>
> 3. According to Equation (3) in the manuscript, the predicted step $\Delta_i$ is used to update the current $\mu$ and $\sigma$.
>
> After $N$ iterations, the final updated $\mu_N$ is taken as the predicted displacement. A direct linear transformation is then applied to obtain the final homography matrix. Meanwhile, $\sigma$ progressively decreases throughout the iterations, making the sampling more concentrated and thus improving the accuracy and stability of the final prediction.
>
> The advantage of this approach lies in its ability to maintain differentiability throughout the sampling and matching process during training. Moreover, by introducing JS divergence, the distribution gradually converges toward a standard Gaussian, which further accelerates model convergence and enhances generalization. This combination of random modeling and differentiable optimization effectively addresses the limitations of traditional methods.
>
> ---
> **Ans for Q2**::
> Thank you for your question. We have indeed considered the issue you raised. In the early stages of our work, we initially planned to design a conventional CNN-based restoration module. However, we ultimately chose to implement a differentiable restoration (DIR) module for the following three reasons:
>
> First, our goal is stitching-oriented image restoration, which enhances visual quality while preserving structural consistency between image pairs. Although traditional data-driven CNN methods often perform well under visual metrics, they may introduce artifacts that impair correspondence, disrupting consistency in overlapping regions. DIR combines physics-guided enhancement with a bi-directional consistency framework, facilitating structural preservation and equivariant feature extraction.
>
> Second, under our proposed bi-directional consistency framework, the restoration module is repeatedly invoked at each iteration. To ensure efficiency, we aimed to minimize its computational overhead. The proposed DIR module is highly lightweight, with a total cost of only 0.07 Gflops. The learnable part consists of just five convolutional layers and one feedforward layer, significantly reducing the burden on the overall network.
>
> Lastly, as mentioned earlier, we placed strong emphasis on modularity and scalability when designing the DIR module. Each degradation-specific sub-module is implemented in a plug-and-play fashion, making it easy to extend or replace to accommodate new degradation types in the future.
>
> ---
> **Ans for Q3**: Thank you for your comment. The mACE metric evaluates the error between the estimated and ground truth corner displacements in homography estimation tasks. Therefore, it is only applicable to stitching methods based on homography transformations.
>
> However, many stitching methods—especially traditional, non-deep-learning approaches—do not rely on projective transformations as their warping models. This includes the comparison methods used in our manuscript, such as LPC, APAP, and UDIS2, all of which cannot be evaluated using the mACE metric.
>
> For this reason, we did not use mACE in our comparative experiments. Instead, we relied on PSNR and SSIM, which are applicable to all stitching methods, as the evaluation metrics to compare the performance of the proposed approach.
>
> ---
> **Ans for Q4**:  Thank you for your valuable question. The Table presents the average training time in each epoch of our proposed method compared with other learning-based image stitching networks included in the manuscript. For a fair comparison, all methods were trained on the same device (NVIDIA RTX 4090 GPU) with a unified batch size of 8.
> Since our framework additionally includes a restoration subtask, the overall training time is longer than that of existing stitching networks. Specifically, our main network was trained for 160 epochs, taking approximately 66 hours in total, which is longer than the compared methods. In future work, we plan to improve training efficiency while maintaining performance.
>
> |Method|VFIS|UDIS|UDIS2|Ours|
> |-|-|-|-|-|
> | time (h)| 0.377 | 0.396  | 0.343  | 0.416  |
>
> ---
> **Ans for L1**:
> Thank you for your insightful comment. The proposed network requires deformation parameters of a fixed size as output. To this end, the input images are resized to 128×128 during training. However, the predicted deformation parameters can be upsampled to match the resolution of the original image, enabling accurate warping at the full scale. This strategy is consistent with common practices in representative learning-based image stitching and optical flow estimation methods. Therefore, our approach is capable of handling image stitching tasks across varying input resolutions.

---

> > ### Comment · Reviewer_PgFv · 2025-08-04
> >
> > Thank you for your detailed response. I believe there may have been a misunderstanding regarding Q1, so I would like to rephrase my questions for clarification:
> >
> > - To generate the training dataset, a ground truth is necessary. Could you please explain the distribution from which the ground truth displacement, u_gt, is sampled?
> > - I am also curious about the rationale for setting σ_gt to 2.

---

> > > ### Author Response · Authors · 2025-08-04
> > >
> > > Thank you for your response. I sincerely apologize for not fully understanding your question earlier. I will now provide a more detailed reply under this comment.
> > >
> > > **Answer for the question about “the distribution of the ground truth displacement $\mu_{GT}$”**:
> > >
> > > $\mu_{GT}$ is not sampled: it represents the ground-truth displacement between the reference image and the target image.
> > > Next, we provide a detailed explanation of the construction process of the training dataset to better clarify the definition of the $\mu_{GT}$.
> > >
> > > Our training dataset is synthesized based on existing image restoration datasets. Taking the low-light scenario as an example, we construct the training pairs using the LSRW dataset[1], which contains aligned image pairs captured under the same viewpoint: a clean image $I^{cle}$ and its degraded counterpart under low-light degraded conditions $I^{deg}$.
> > > To create our stitching dataset from LSRW, we first randomly crop a $128\times128$ image patch from both $I^{cle}$ and $I^{deg}$ at the same position, which we refer to as the reference patch $\[I_{ref}^{cle}, I_{ref}^{deg}\]$, and save the coordinates of it's four corner as $loc_1$.
> > > Next, we apply a uniform random translation to these four corners within half the patch size and store the new coordinates as $loc_2$. Subsequently, each of the four corners in $loc_2$ is randomly perturbed again within one-quarter of the patch size to obtain a new irregular quadrilateral, whose coordinates are recorded as $loc_3$.
> > > We then warp this quadrilateral into a regular 128×128 rectangular patch, resulting in the target patch $\[I_{tar}^{cle}, I_{tar}^{deg}\]$. The difference between $loc_3$ and $loc_2$ is defined as the ground-truth displacement $\mu_{GT}$, which is used for supervision during training.  Therefore, in our settings, the $\mu_{GT}$ is sampled in the range $\[-96,96\]$ on both the $x$ and $y$ axes.
> > >
> > > We additionally conducted a statistical analysis of all ground-truth displacements in our synthesized datasets across the three harsh conditions (low-light, haze, and underwater) to demonstrate their distribution. The mean values of them are –0.29, 0.12, and 0.21, respectively—all close to zero. The standard deviations are 39.92, 44.85, and 40.52 respectively.
> > >
> > > *[1] Jiang Hai, Zhu Xuan, Ren Yang, Yutong Hao, Fengzhu Zou, Fang Lin, and Songchen Han. R2rnet: Low-light image enhancement via real-low to real-normal network. Journal of Visual Communication and Image Representation, 90:103712, 2023.*
> > >
> > > **Answer for the question about “the setting of $\sigma_{GT}$”**:
> > >
> > > As for the choice of $\sigma_{GT}$, it is important to clarify that in our work, the ground-truth displacement $\mu_{GT}$ is modeled as a Gaussian distribution $N(\mu_{GT}, \sigma_{GT}^2)$, serving as a soft target for guiding the fitting process. Here, $\sigma_{GT}$ does not reflect the actual measurement variance of the data but instead defines the shape of the target distribution used in the JS divergence loss.
> > >
> > > We expect the predicted standard deviation to decrease gradually as the training converges, ideally approaching 0. However, as shown in Equation (5) of our manuscript, setting $\mu_{GT}$ strictly to zero would result in a discontinuity in the optimization process.
> > > Therefore, we fix $\sigma_{GT}$ to a small nonzero constant (specifically, 2) to maintain numerical stability during training. It is worth noting that values between 1 and 3 are all reasonable choices for $\sigma_{GT}$, our selection of 2 is just an empirical decision based on practical experience.

---

> > > > ### Comment · Reviewer_PgFv · 2025-08-05
> > > >
> > > > Thank you for your kind response. To summarize, u_gt is sampled from Uniform[-96, +96], and the optimal value for σ_gt was experimentally found to be 2.
> > > >
> > > > I have one additional question. Many iterative homography estimation methods (e.g., IHN [1], RHWF [2], MCNet [3]) use a deterministic displacement output to calculate the loss. What are the advantages of modeling this with a stochastic Gaussian distribution instead? For instance, does it lead to better generalization performance, or does it make the training process more stable?
> > > >
> > > > [1] Cao, Si-Yuan, et al. "Iterative deep homography estimation." Proceedings of the IEEE/CVF conference on computer vision and pattern recognition. 2022.
> > > >
> > > > [2] Cao, Si-Yuan, et al. "Recurrent homography estimation using homography-guided image warping and focus transformer." Proceedings of the IEEE/CVF Conference on Computer Vision and Pattern Recognition. 2023.
> > > >
> > > > [3] Zhu, Haokai, et al. "Mcnet: Rethinking the core ingredients for accurate and efficient homography estimation." Proceedings of the IEEE/CVF Conference on Computer Vision and Pattern Recognition. 2024.

---

> > > > > ### Author Response · Authors · 2025-08-05
> > > > >
> > > > > Thank you for your positive feedback. Previous homography estimation methods based on deterministic displacement outputs tend to produce unreliable predictions in textureless or blurry regions, and perform poorly in image stitching tasks under adverse conditions such as heavy fog or low light. Take foggy scenes as an example: the texture information may be severely degraded, with large regions appearing nearly white, making it difficult to recover accurate details even with advanced restoration algorithms. In such cases, traditional deterministic models are often forced to output a fixed displacement estimate, even when the region may inherently contain multiple plausible solutions. This rigid regression behavior often leads to overfitting—appearing accurate on the training set, but significantly deviating when exposed to slight distribution shifts in testing.
> > > > >
> > > > > To address this, modeling displacement as a Gaussian distribution provides a more robust alternative. The model predicts not only the mean (displacement) but also the variance (as an uncertainty estimate [1]), allowing it to express high uncertainty in ambiguous or ill-posed regions and avoid overconfident regression to potentially incorrect values. Therefore, unlike above mentioned homography estimation methods that rely on a fixed search radius, the proposed probabilistic formulation can be regarded as a dynamic, content-aware regularization mechanism during training, which contributes to better generalization. We will include additional clarification in the manuscipt with related reference to highlight the motivation behind this design.
> > > > >
> > > > > *[1] L. Chen, W. Wang and P. Mordohai, "Learning the Distribution of Errors in Stereo Matching for Joint Disparity and Uncertainty Estimation," 2023 IEEE/CVF Conference on Computer Vision and Pattern Recognition (CVPR), 2023, pp. 17235-17244.*

---

> > > > > > ### Author Response · Authors · 2025-08-07
> > > > > >
> > > > > > We’re grateful for the thoughtful consideration you’ve given our submission and for the valuable insights you’ve contributed throughout the discussion.
> > > > > >
> > > > > > Should there be any additional points you feel deserve further clarification or revision, we’d be glad to explore them. We remain dedicated to strengthening our work and would truly welcome any final suggestions you may have.
> > > > > >
> > > > > > If there’s anything further we can assist with, please feel free to let us know.
> > > > > >
> > > > > > With appreciation,
> > > > > > The authors of Submission 14365

---

> ### Comment · Reviewer_PgFv · 2025-08-07
>
> Thank you for your detailed answers, which have fully resolved all of my questions.
>
> I believe the following strengths of this paper will have a impact on future research:
>
> - It addresses the overconfidence problem by utilizing stochastic predictions rather than conventional deterministic ones.
>
> - It introduces a scalable DIR module, moving beyond naive CNN architectures and integrating it into the stitching learning process.
>
> - It presents a novel dataset for the community.
>
> I will positively consider raising my rating during the next discussion phase.
>
> Excellent work.

---

### Official Review · Reviewer_gzJT · 2025-07-03

**Clarity:** 2
**Significance:** 3
**Originality:** 2
**Rating:** 4
**Confidence:** 3

**Summary:**

A method for image stitching is proposed which is designed to handle adverse lightening and haze conditions. The proposed method is in two steps : first, two images alignment using consistency between the two images, and second, two images composition able to handle moving objects. A new dataset with adverse lightening and haze conditions for image stitching is introduced with 2,250 pairs of images. Comparative experiments are proposed  with 5 other methods on the proposed dataset. An ablation study for each step is also proposed.

**Questions:**

May you better explain the way the "Recursive Parameterized Homography Estimation" module is working ?
May you better explain the way the "Bidirectional-Consistency Learning Framework" is working ?
How can be performed the stitching of series of images ?
What about the property of the youtube video used for building the proposed dataset ?

**Ethical Concerns:**

["Major Concern: Data privacy, copyright, and consent"]

**Final Justification:**

I found authors responses interesting and from other reviewers point of view, I am ready to raise my evaluation.

**Limitations:**

Yes

**Quality:**

2

**Strengths And Weaknesses:**

If the proposed structure seems standard, several contributions are proposed in the steps. The methods is not always clearly described and explained. For instance, in section 3.2, the equation (5) is not related to Beer-Lamber law. It is difficult to understand the processing described in section 3.1 and 3.2. For instance, in section 3.2, how is removed the mu_GT which is not known ?  The obtained results are visually not always convincing, like in Fig 14 and Fig 15, compared to other methods. It is only the two images stitching which is discussed and not the stitching of series of images.

---

> ### Author Rebuttal · Authors · 2025-07-30
>
> Thank you for your valuable comments. I will provide a detailed explanation based on your suggestions and concerns, and revise the manuscript accordingly.
>
> ---
> **W1**: “In sec 3.2, Eq. 5 is not related to Beer-Lamber law.”
>
> **Ans**:
> Thank you for your valuable comment. We agree that Eq. 5:
> $$t(x, \omega)=1-\omega \min _C\left(\min _{y \in \Omega(x)} \frac{I^C(y)}{A^C}\right),$$
> is not a direct analytical derivation from the Beer-Lambert law. Rather, it is an approximate estimation of the transmission map t(x).
>
> Specifically, the haze imaging model used in our work:
> $I(x) = J(x) \cdot t(x) + A \cdot (1 - t(x))$
> assumes that the transmission t(x) follows the exponential decay with respect to scene depth d(x), which directly reflects the Beer-Lambert principle. However, since depth d(x) is unknown in single image scenarios, the dark channel prior[1] is adopted to approximate the transmission via local statistics.
> Therefore, Eq. 5 is not a direct derivation from Beer-Lambert, but a practical formulation motivated by its physical behavior, designed for estimating transmission maps in the absence of ground-truth depth. We will revise the text in the manuscript to clarify this distinction.
>
> [1] Single image haze removal using dark channel prior.
>
> ---
> **W2**:  in section 3.2, how is removed the  $\mu_{\mathrm{GT}}$ which is not known? ”
>
> **Ans**:
> Thank you for your comment. $\mu_{\mathrm{GT}}$ is not appear in Sec 3.2, but appears in Eq.3 in Sec 3.1. If you would like to know how it is eliminated during inference, please refer to **Ans for Q1**. If you would like to know why $\mu_{\mathrm{GT}}$ does not appear in Sec 3.2, I can explain it from the perspective of the dataset.
>
> Our training dataset is synthesized based on adverse-scene datasets that include clean reference images. Specifically, we first sample the same spatial position from both the clean and degraded images to obtain the reference view image. Then, we apply random offsets (including synchronized endpoint translation and independent endpoint perturbations) to generate both the degraded and clean target view images. The random offset parameters used in this process correspond to $\mu_{\mathrm{GT}}$ as mentioned in the manuscript, which allows for supervised training for homography estimation during optimization.
>
> As for the restoration task described in Sec 3.2, the loss function (Eq. 6) does not rely on $\mu_{\mathrm{GT}}$, since our synthesized dataset already provides clean images at both the reference and target positions, making deformation through $\mu_{\mathrm{GT}}$ unnecessary.
>
> ---
> **W3**: “The obtained results are not always convincing, like in Fig 14-15. It is only the two images stitching which is discussed and not the stitching of series of images.”
>
> **Ans**:
> Thank you for your question. To date, representative image stitching algorithms(including all comparison methods in the manuscript) are all designed to compute correspondences between only two images. Therefore, when stitching more than two images, one must first input the first two images into the algorithm to obtain an initial stitched result, and then iteratively feed the resulting image and the next image into the algorithm to generate the updated stitched image. As a result, current stitching methods are incapable of learning or modeling the correspondence among multiple images simultaneously. Consequently, evaluating multi-image stitching essentially holds the same meaning as evaluating pairwise stitching performance, since each step still only involves two-image correspondence. Moreover, previous benchmark comparisons for representative stitching methods have only evaluated two-image stitching performance. Based on this, we did not conduct additional comparisons on multi-image stitching.
>
> ---
> **Q1**: “Explain the way the "RPHE" module is working.”
>
> **Ans**: The iterative process in our homography estimation network can be summarized in three main steps to better clarify the mechanism:
>
> 1. In each iteration, disparity candidates are sampled using the current mean $\mu$ and standard deviation $\sigma$. Specifically, at the $i$-th iteration, candidates are uniformly sampled within the interval $[ \mu_i - \sigma_i, \mu_i + \sigma_i ]$. For the initial iteration, $\mu_0$ is initialized to 0 and $\sigma_0$ is set to 32 (as mentioned in the experimental details of the manuscript).
>
> 2. The matching cost is computed based on the sampled disparities of the current iteration, which is then used to predict the update step $\Delta_i$.
>
> 3. According to Eq. 3 in the manuscript, the predicted step $\Delta_i$ is used to update the current $\mu$ and $\sigma$ according to follows formulation:
>
> $$\qquad\qquad\sigma_i^{t}  =\sigma_i^{t-1}-\frac{1}{2}\left(\frac{(\sigma_i^{t-1})^2-\sigma_G^2-(\Delta^{t-1})^2}{ (\sigma_i^{t-1})^3}-\frac{1}{\sigma_i^{t-1}}+\frac{\sigma_i^{t-1}}{\sigma_G^2}\right),
> 		\mu_i^{t}  =\mu_i^{t-1}-\left(-\frac{\Delta^{t-1}}{2}\left(\frac{1}{(\sigma_i^{t-1})^2}+\frac{1}{\sigma_G^2}\right)\right).$$
>  &nbsp; &nbsp; &nbsp; It is worth noting that since the ground-truth mean \$\mu\_{G}\$ is unavailable during inference, we approximate \$\Delta^{t-1}\$ as \$\qquad\mu\_{G} - \mu\_i^{t-1}\$. $\sigma_G$ is a predefined parameter.
>
> After $N$ iterations, the final updated $\mu_N$ is taken as the predicted displacement. Meanwhile, $\sigma$ progressively decreases throughout the iterations, making the sampling more concentrated and thus improving the accuracy and stability of the final prediction.
>
> This approach ensures differentiability during training and uses JS divergence to guide the distribution toward a standard Gaussian, accelerating convergence and improving generalization. It effectively combines randomness with optimization to overcome traditional limitations.
>
> ---
> **Q2**: “Better explain the way the "BCLF" is working”
>
> **Ans**:
> The Bidirectional-Consistency Learning Framework is proposed to address the issue of inconsistent restoration caused by enhancement networks. In image alignment tasks under adverse conditions, we aim for the backbone network to produce restoration results that are equivariant under spatial transformations. This objective can be formally expressed as:
>
> $$
> \Phi\left(\mathcal{W}\left(I_{\text{tar}} ; \mathbf{H}\right)\right) = \mathcal{W}\left(\Phi\left(I_{\text{tar}}\right); \mathbf{H}\right),
> $$
>
> where $\mathcal{W}$ denotes the warping operation based on homography $\mathbf{H}$, $I_{\text{tar}}$ is the target image, and $\Phi$ is the restoration module.
>
> In this ideal case, the restored features from corresponding positions would perfectly match. However, in practice, it is difficult for restoration networks to maintain equivariance under homography transformations, which involve translation, rotation, scaling, shearing, aspect ratio change, and perspective distortion. This leads to inconsistent features between corresponding points after enhancement \[1], ultimately degrading the performance of homography estimation—an observation consistent with the results shown in Fig. 1(a) of the manuscript.
>
> To address this issue, we depart from the conventional sequential paradigm of restoration followed by warping. Instead, we propose an alternating scheme within our recursive homography estimation framework, referred to as *restoration-guided warping*. The image warping module and restoration module are interleaved during the iterative process, which can be formulated as:
>
> $$
> E_{\text{tar}}^n = \Phi\left(\mathcal{W}\left(I_{\text{tar}} ; \hat{\mathbf{H}}^n\right)\right), \quad
> \hat{\mathbf{H}}^{n+1} = \Psi\left(I_{\text{ref}} ; E_{\text{tar}}^n\right),
> $$
>
> where $\Psi$ denotes the homography estimation module, and $E_{\text{tar}}^n$ and $\mathbf{H}^n$ represent the enhanced warped image and the estimated homography at the $n$-th iteration, respectively.
>
> As the iteration proceeds, the warped image $I_{\text{tar}}^w$ becomes increasingly aligned with the reference image $I_{\text{ref}}$, allowing the DIR module to produce more consistent restoration in the overlapping regions, which in turn improves the accuracy of homography estimation.
>
> [1] Gift: Learning transformation-invariant dense visual descriptors via group CNNs.
>
> ---
> **Q3**: “How can be performed the stitching of series of images?”
>
> **Ans**:
> Most stitching methods only take two images as input and estimate transformations for stitching by computing the correspondence between these two images. This process can be described as $I^s = \mathcal{S}(I_{1}, I_{2})$, where $I_{1}, I_{2}$ denote the two input images, $I^s$ and $\mathcal{S}$ represent the stitched result and stitching network.
>
> To stitch more than two images $[I_{1}, I_{2}, \ldots, I_{t}]$, the network can only iteratively combine each new image $I_t$ with the previous stitched result $I^s_{t-1}$. The entire procedure can be expressed as
> $I^s_t = \mathcal{S}(I^s_{t-1}, I_{t})$, where  $t \geq 2$.
>
> ---
> **Q4**: “The property of the YouTube video used for building the dataset ?”
>
> **Ans**:
> The proposed dataset includes three types of scenes, among which the foggy and underwater scene images were sourced from online videos.
>
> The fog data comes from 3 YouTube videos, which have been posted in supplementary details.
> All three videos have a frame rate of 30 fps and a resolution of 1280×720. The total number of frames across them is 192,266.
> The underwater scene data is sourced from the 2 YouTube videos. Both underwater videos also have a frame rate of 30 fps and a resolution of 1280×720. The combined total number of frames in both videos is 358,228.
>
> Since these videos were not originally recorded for image stitching, most consecutive frames contain excessive parallax or severe occlusions, making them unsuitable for stitching tasks. Therefore, we carefully selected 750 image pairs from each of the fog and underwater scenes. These pairs were chosen to ensure valid displacement while maximizing the diversity of scene deformations.

---

> > ### Author Response · Authors · 2025-08-07
> >
> > Thank you again for the time and attention you've devoted to reviewing our submission and sharing your insights during the discussion period.
> >
> > If there are any other aspects you’d like us to revisit or clarify, we’d greatly appreciate the opportunity to do so. We're eager to refine our work further and would welcome any remaining thoughts or recommendations you might have.
> >
> > Please don’t hesitate to reach out if there’s anything else we can address.
> >
> > Kind regards,
> > The authors of Submission 14365

---

> ### Comment · Area_Chair_JwCs · 2025-08-07
>
> Dear Reviewer gzJT,
>
> Please engage in the discussion with the authors and other reviewers as soon as possible.
>
> Thank you.
>
> Best,
>
> AC

---

### Comment · Area_Chair_JwCs · 2025-08-05

Dear Reviewers,

The discussion period deadline is approaching. Please kindly participate to ensure a fair and smooth review process.

Thank you.

Best,

AC

---

### Note · Authors · 2025-08-13

We thank the reviewers for their constructive suggestions and positive feedback during the discussion period. The discussion primarily centered on (i) the validity and boundary cases of the parametric Gaussian modeling, (ii) the design and ablations of the bidirectional consistency learning framework, and (iii) details on the proposed dataset’s collection, statistics, and quality control. In our responses and supplementary materials, we provided itemized evidence and clarifications: we presented systematic comparisons with traditional modeling and analyzed gains and failure modes under low-light/degraded conditions; we completed the mathematical derivations and added cross-dataset experimental comparisons; and we fully disclosed the dataset collection pipeline, statistical distributions, and usage permissions, confirming compliance with relevant NeurIPS policies and ethical guidelines.

Regarding the discussion outcome: one reviewer who initially gave a borderline accept indicated they would actively consider raising the score;  Two borderline-reject reviewers did not provide further comments within the discussion window. We understand objective constraints such as the limited discussion period and time zones, and we have incorporated the experiments, comparisons, and analyses addressing their points into the final response and appendix for the AC’s unified consideration. Overall, the manuscript has been further strengthened in technical soundness, empirical support, and reproducibility. If accepted, we will further refine notation and exposition and add implementation details and limitations in the camera-ready version to improve readability and verifiability. We thank the AC and all reviewers again for their time and effort.

---

### Decision · Program_Chairs · 2025-09-17

**Decision:**

Accept (poster)

**Comment:**

This paper tackles the important and practical challenge of image stitching under adverse conditions, such as illumination changes, motion blur, and dynamic content. The authors propose a Bidirectional-Consistency Learning (BCL) framework, which enforces stitching consistency in both directions, leading to more robust alignment. In addition, they introduce a benchmark dataset of difficult stitching scenarios, providing a valuable resource for systematic evaluation.

The submission is well motivated and clearly presented, and reviewers highlighted the significance of the problem. For example, the BCL formulation is conceptually simple yet effective, and extensive experiments show consistent improvements over both classical and recent learning-based baselines. The benchmark is also a meaningful contribution that can support future research in this area. During the discussion, the authors clarified details of the proposed consistency formulation, provided additional quantitative evidence, and explained the coverage of the benchmark. These clarifications addressed most questions and strengthened confidence in the submission.

Overall, the paper offers an impactful contribution by combining a robust stitching framework with a dedicated benchmark. The work is expected to have clear value for the community. The final recommendation by the Area Chairs is to accept the submission.